# Helium in diamonds unravels over a billion years of craton metasomatism

Yaakov Weiss [1,2✉], Yael Kiro[3,2], Cornelia Class[2], Gisela Winckler [2,4], Jeff W. Harris[5] & Steven L. Goldstein[2,4]

Chemical events involving deep carbon- and water-rich fluids impact the continental lithosphere over its history. Diamonds are a by-product of such episodic fluid infiltrations, and entrapment of these fluids as microinclusions in lithospheric diamonds provide unique opportunities to investigate their nature. However, until now, direct constraints on the timing of such events have not been available. Here we report three alteration events in the southwest Kaapvaal lithosphere using U-Th-He geochronology of fluid-bearing diamonds, and constrain the upper limit of He diffusivity (to $D \approx 1.8 \times 10^{-19}$ cm$^2$ s$^{-1}$), thus providing a means to directly place both upper and lower age limits on these alteration episodes. The youngest, during the Cretaceous, involved highly saline fluids, indicating a relationship with late-Mesozoic kimberlite eruptions. Remnants of two preceding events, by a Paleozoic silicic fluid and a Proterozoic carbonatitic fluid, are also encapsulated in Kaapvaal diamonds and are likely coeval with major surface tectonic events (e.g. the Damara and Namaqua–Natal orogenies).

[1] The Freddy and Nadine Herrmann Institute of Earth Sciences, The Hebrew University of Jerusalem, Jerusalem, Israel. [2] Lamont-Doherty Earth Observatory of Columbia University, Palisades, NY, USA. [3] Department of Earth and Planetary Sciences, Weizmann Institute of Science, Rehovot, Israel. [4] Department of Earth and Environmental Sciences, Columbia University, Palisades, NY, USA. [5] School of Geographical and Earth Sciences, University of Glasgow, Glasgow, UK. ✉email: yakov.weiss@mail.huji.ac.il

The continental lithospheric mantle (CLM) has been che-mically altered by infiltrating carbon- and water-rich (C-O-H) fluids throughout its history[1–3]. This process, generally referred to as "mantle metasomatism," is prevalent and involves enrichment of the CLM in volatiles and highly incompatible trace elements, as reflected by the mineralogy and chemical composition of CLM samples (xenoliths and xenocrysts) and their host alkaline magmas[4–9]. Constraining the timing of these events and the nature of the fluids involved is an ongoing challenge for our understanding of the CLM evolution and history, including its long-term chemical and physical properties[10–13].

Diamonds are a primary target for studies of deep C-O-H fluids since metasomatic interaction between such fluids and CLM rocks often leads to their formation[14–16]. Diamonds grow from carbon supplied by the fluid[14,17,18] and thus their crystallization ages represent the timing of CLM metasomatic events. However, there is no technique available to date diamonds directly because the diamond lattice does not contain sufficient quantities of any radioisotope. As a result, current approaches use radiogenic isotope dating of silicates and sulfide inclusions to obtain the age of their host diamonds and the metasomatic events in which they grew[19–23]. The logic is based on the key assumption that diamonds and the inclusions they encapsulate are syngenetic (i.e., formed simultaneously). This idea is supported by observations that the morphology of included minerals are commonly imposed by the diamond[24–27]. On the other hand, increasing evidence points to the possible protogenetic origin of inclusions (pre-existing minerals incorporated during diamond crystallization) that has generated a debate about the significance of the information recorded by inclusions for the age of their host diamonds[28–31]. Nevertheless, it is also possible that inclusions are protogenetic but still record the age of diamond formation, provided that the inclusion and the diamond-forming fluid reached equilibrium during the metasomatic event[32].

Some diamonds encapsulate direct samples of deep C-O-H mantle fluids in the form of high-density supercritical fluid (HDF) microinclusions, which vary between saline, silicic, and carbonatitic compositions[33–35]. The potential of dating these HDF-bearing diamonds would provide a means to circumvent the assumptions concerning diamond age determination from mineral inclusions. In addition, HDFs offer a unique record on the compositions of the fluid involved in the metasomatic event[35,36], whereas mineral inclusions only allow the composition to be indirectly inferred from geochemical proxies or modeling based on mineral/melt partition coefficients. To date, however, the limited available Sr-Nd-Pb isotopic data on HDF-bearing diamonds do not offer a straightforward age significance[35–37]. Thus, many studies have used the temperature and time dependency of nitrogen atom aggregation in the diamond lattice to place temporal constraints on the formation of HDF-bearing diamonds. But this method has limitations as well, since nitrogen aggregation is more sensitive to temperature than time, resulting in variable diamond formation ages between a few kilo-years and billions of years as a function of possible temperature histories between the time of diamond formation and its ascent to the surface by kimberlite eruption[38,39]. Moreover, along a cratonic geotherm, variable nitrogen aggregation states in diamonds from the same locality do not necessarily denote different ages, rather the nitrogen aggregation variability could indicate derivation from different depths and therefore reflect differences in temperature.

Recently, a different approach has been suggested for constraining the timing of HDF-bearing diamond formation and C-O-H metasomatism by combining He isotope analyses with U-Th-He abundance measurements[40,41]. Like (U-Th)/He chronometry of other minerals[42,43], this approach is based on the accumulation of $^4$He atoms produced by α-decay of U and Th, while at the same time it considers all possible sources contributing to the budget of

He in a diamond. For example, it accounts for the presence of significant initial $^4$He in many C-O-H fluid-bearing diamonds. This concept has been investigated in a study of alluvial HDF-bearing diamonds[41], which showed a positive correlation between $^4$He and U-Th concentrations, reflecting the low diffusivity of He in diamond at mantle temperatures[44–47]. However, that study did not take into account the possibility of some diffusive loss of He, which is critical for a meaningful interpretation of (U-Th)/He data and for constraining crystallization ages[43].

This study reveals strong relationships between helium concentrations, $^4$He/$^3$He ratios, (U-Th)/He ratios, and the nitrogen aggregation characteristics of ten HDF-bearing diamonds from the De Beers Pool and Finsch mines in the southwest Kaapvaal craton, South Africa. Taking into account the impact of possible He diffusion on the diamonds $^3$He/$^4$He ratios with time (previously estimated to fall within a large range between $D = 10^{-16}–10^{-21}$ cm$^2$ s$^{-1}$ [44–47]), and considering the thermal and tectonic history of the Kaapvaal craton, we suggest an upper limit of ~$1.8 ± 0.2 × 10^{-19}$ cm$^2$ s$^{-1}$ for the diffusivity of He in HDF-bearing diamonds. This diffusion limit provides a major step forward for constraining diamond crystallization ages based on U-Th-He geochronology, and for unraveling the timing of C-O-H metasomatic events in the context of regional tectonics and volcanism. For the diamonds analyzed in the present study, it constrains the timing of crystallization during three alteration events, each by a different fluid agent. The youngest episode, by highly saline fluids, was coeval to the Kaapvaal late-Mesozoic kimberlite eruptions, and silicic and carbonatitic fluids were responsible for preceding metasomatic events that took place during the Paleozoic and Proterozoic.

## Results

### Helium in HDF-bearing diamonds from the southwest Kaapvaal craton

We report helium contents and isotopic compositions of ten HDF-bearing diamonds from De Beers Pool and Finsch mines (Supplementary Data 1[48]) that were previously analyzed for nitrogen aggregation and HDFs' major- and trace-element compositions[49,50]. The South African HDF-bearing diamonds have low $^3$He/$^4$He ratios (Fig. 1a) compared to both mid-ocean-ridge basalts (MORB, with $^3$He/$^4$He $= 8 ± 1$ Ra; where Ra is the atmospheric ratio of $1.39 × 10^{-6}$) and ocean island basalts (OIBs, $^3$He/$^4$He $= 5–50$ Ra), including the so-called "low-$^3$He/$^4$He" OIBs ($^3$He/$^4$He $< 7$ Ra)[51,52]. Rather, their $^3$He/$^4$He ratios are characterized by Ra values that range between MORB and continental crust (CC, $^3$He/$^4$He $< 1$ Ra)[53] (Fig. 1a, b). A strong connection is observed between the measured $^3$He/$^4$He ratios, the HDFs' major-element compositional type (saline, silicic, or high-Mg carbonatitic), and the diamond nitrogen aggregation characteristics (Figs. 1 and 2a). The saline HDF-bearing diamonds are characterized by $^3$He/$^4$He ratios of 2.7–4.4 Ra ($^4$He/$^3$He $= 163–272 × 10^3$), whereas the diamonds containing silicic and high-Mg carbonatitic HDFs have much lower $^3$He/$^4$He ratios between 0.07 and 0.69 Ra ($^4$He/$^3$He $= 1000–11,000 × 10^3$). At the same time, the saline HDF-bearing diamonds analyzed here contain nitrogen only in A centers (nitrogen atom pairs replacing two adjacent carbon atoms), whereas diamonds of silicic and high-Mg carbonatitic HDF compositions contain nitrogen in both A and B centers (with 25–35% in B centers, where four nitrogen atoms and an atomic vacancy substitute for five carbon atoms). Available literature data for seven additional HDF-bearing diamonds from De Beers Pool and two from the neighboring Koffiefontein mine[54] are overall consistent with this observation; all are characterized by $^3$He/$^4$He $= 3.4–5.1$ Ra ($^4$He/$^3$He $= 143–214 × 10^3$), nitrogen is solely in A centers, and nearly all (eight of nine) have saline HDF compositions (Figs. 1 and 2a). The new data also show a strong positive relationship between $^4$He/$^3$He and (U + Th)/He ratios (Fig. 2b), which is expressed on a

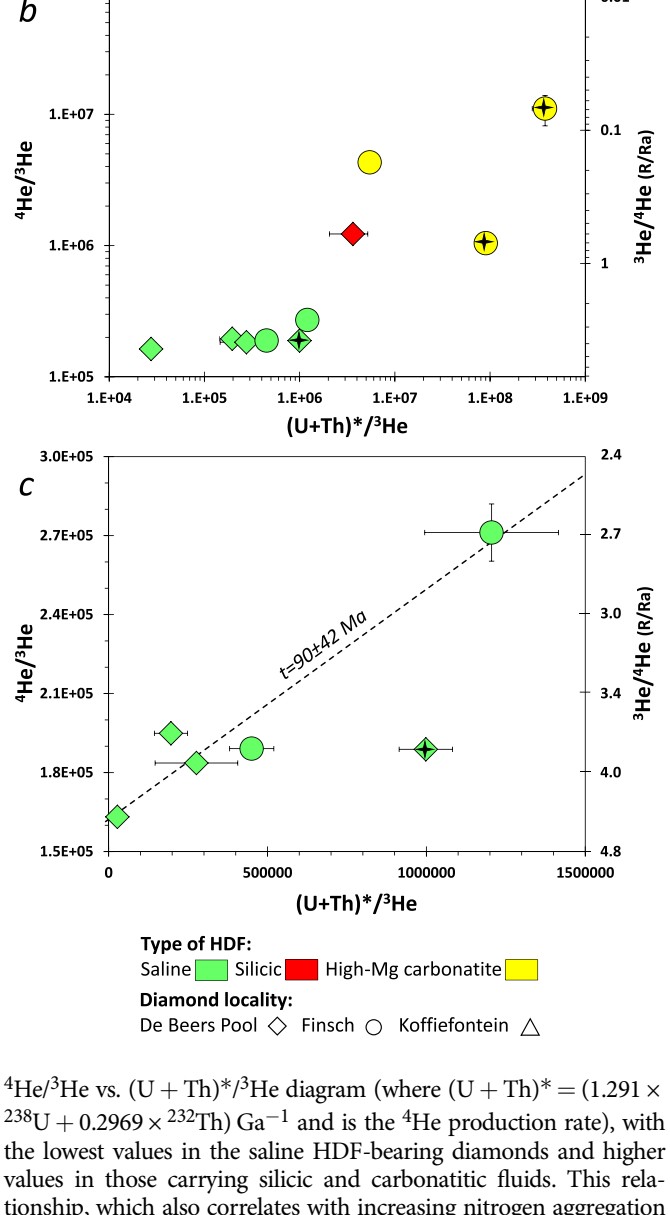

**Fig. 1 Helium content and $^3$He/$^4$He ratios of HDF-bearing diamonds from De Beers Pool and Finsch mines.** Compositional type of high-density supercritical fluid (HDF) microinclusions and diamond mine locality are coded by color and shape, respectively. Additional literature data (small symbols) for De Beers Pool and two diamonds from the neighboring Koffiefontein mine[54] are also shown. The $^4$He content and $^3$He/$^4$He errors represent standard deviation and are smaller than the size of the symbols for the majority of data points. For comparison we plot: **a** The $^3$He/$^4$He range for mid-ocean ridge basalts (MORB)[51], ocean island basalts (OIB)[52], continental crust (CC)[53], and the values for a gas-rich MORB, the "Popping Rock"[99]. With time $^3$He/$^4$He ratios decrease due to U-Th decay (indicated by the shaded arrow left of the y-axis). **b** Published He content and $^3$He/$^4$He for continental lithospheric mantle (CLM)-derived xenoliths, and their average and median $^3$He/$^4$He values (only data determined by crushing analyses are presented, as opposed to heating and fusion; dashed lines represent the ±standard deviation on the average; Supplementary Data 2[69]). Symbols for Kaapvaal craton xenoliths are distinguished from other CLM provinces; among these, xenoliths with late-Mesozoic emplacement ages (Mz) show $^3$He/$^4$He mostly between 5.5 and 4.5 Ra, whereas xenoliths of Proterozoic (ρ) age (1.2 Ga) are characterized by radiogenic <1 Ra values. Also shown is the range of $^3$He/$^4$He ratio of the Wesselton kimberlite from De Beers Pool[81]. All $^3$He/$^4$He values are normalized to the atmospheric ratio Ra = $1.39 \times 10^{-6}$.

$^4$He/$^3$He vs. (U + Th)*/$^3$He diagram (where (U + Th)* = (1.291 × $^{238}$U + 0.2969 × $^{232}$Th) Ga$^{-1}$ and is the $^4$He production rate), with the lowest values in the saline HDF-bearing diamonds and higher values in those carrying silicic and carbonatitic fluids. This relationship, which also correlates with increasing nitrogen aggregation in the host diamonds, reflects the ingrowth of radiogenic $^4$He with time since diamond formation at depth. Moreover, excluding one

**Fig. 2** $^3$He/$^4$He (and $^4$He/$^3$He) versus nitrogen aggregation, and (U+Th)*/$^3$He in HDF-bearing diamonds from the southwest Kaapvaal craton. (U + Th)* = (1.291 × $^{238}$U + 0.2969 × $^{232}$Th) Ga$^{-1}$ is the $^4$He production rate. **a** Relationships between HDF compositional type, $^3$He/$^4$He ratio, and host diamond nitrogen aggregation indicate formation from different metasomatic events throughout the Kaapvaal CLM history. The %B error bars signify 5% and are a maximum estimate. For diamonds with %B = 0 it is represented by the gray square; the range of $^4$He/$^3$He ratios also is expanded for clarity. Error bars for $^3$He/$^4$He and $^4$He/$^3$He ratios are standard deviations. **b** $^4$He/$^3$He vs. (U + Th)*/$^3$He for all the samples; the silicic and carbonatitic samples show both higher $^4$He production rates and higher $^3$He/$^4$He than the saline ones. Uncertainties on (U + Th)*/$^3$He are propagated based on the U, Th, and $^3$He errors (Supplementary Data 1[48]). **c** Same plot as in (**b**) showing only the saline HDF-bearing diamonds and the errorchron indicating diamond formation and metasomatism at 90 ± 42 Ma; systematics is in "Methods—'U-Th-He isochron ages'," and ages and errors are calculated using IsoplotR[87]. HDF compositional type and mine symbols are as in Fig. 1, data points with stars in (**b**, **c**) are two carbonatitic diamonds with internal microinclusion-bearing and overgrown microinclusion-free zones ("cloudy diamonds")[50] and a saline diamond with unusually high U/Th (i.e., Th/U < 1)[49]. These diamonds show apparent high (U-Th)*/He ratios and were not used for age determination (additional info is in the section "The significance of microinclusion homogeneity").

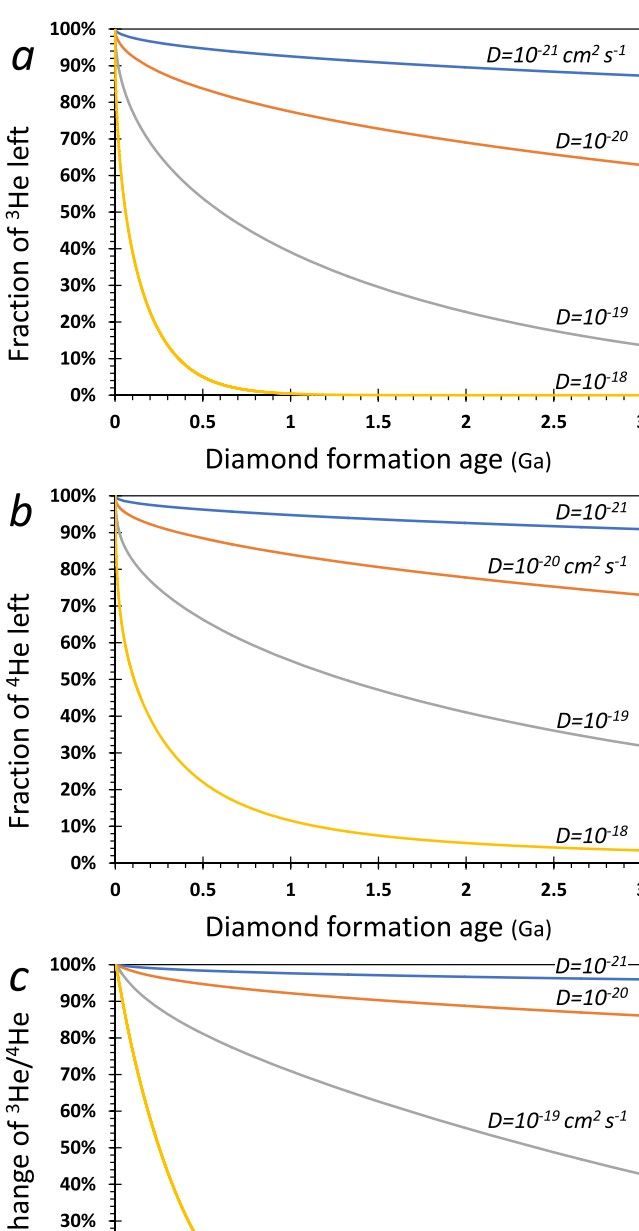

**Fig. 3 Effects of diffusion on He isotopes.** The fraction of initial He content (**a**, **b**) and the fractional change in $^3$He/$^4$He (**c**) in a diamond as a function of its formation age for different He diffusivities. Model results take into account radioactive production of $^4$He and removal of $^3$He and $^4$He by diffusion (model details are in "Methods—'Diffusion model of He in diamonds'"). **a** The remaining fraction (in %) of $^3$He compared to its initial amount, which remains constant over time for the case of no diffusion. **b** The percent remaining of $^4$He compared to its evolved amount (initial + radiogenic-formed) assuming no diffusion. **c** The fractional change of the $^3$He/$^4$He ratio due to diffusion compared to its evolved value for no diffusion. The changes over time show that given any age, if $D$ is larger the $^3$He/$^4$He ratio is lower. Each curve in (**a**–**c**) corresponds to the time-integrated change of a single panel in Fig. S1.

diamond with an unusually high U/Th ratio (i.e., with Th/U < 1)[49], the saline HDF-bearing diamonds' $^4$He/$^3$He vs. (U + Th)*/$^3$He defines a positive trend, which, if interpreted as an age, corresponds to 90 ± 42 Ma and an initial $^4$He/$^3$He ratio of 162 ± 25 × 10$^3$ ($^3$He/$^4$He = 3.9–5.3 Ra; Fig. 2c; explained in "Methods—'U-Th-He isochron ages'"). This apparent age is in agreement with the low-aggregated nitrogen of these HDF-bearing diamonds and the eruption ages of their host kimberlites (De Beers Pool—85 ± 5 Ma, Finsch—118 ± 3 Ma, e.g., refs. [55–57]; the Finsch host volcanic is an olivine lamproite, formerly Type 2 kimberlite or orangeite, but we are using the generalized term kimberlite throughout the manuscript).

**Diffusion of helium in HDF-bearing diamonds.** The prime source of He in HDF-bearing diamonds is the microinclusions[41,58,59]. Additional possible sources of He from the diamond lattice are negligible, or can be avoided by crushing release of He from an inner fragment of deep mined diamonds (as discussed in detail by Timmerman et al. [41], and here in the Supplementary information—"The budget of He in HDF-bearing diamonds"). Timmerman et al. [41] also assumed that He diffusion is slow and insignificant; but considering the large range of estimated He diffusivities of between $D = 10^{-16}–10^{-21}$ cm$^2$ s$^{-1}$ [44–47], this process may impact the diamond's potential to retain He over geological time scales of hundreds of millions to billions of years, influencing the estimated age. To determine the possible effects of He loss by diffusion on the $^3$He/$^4$He ratios and (U-Th)/He age determinations of HDF-bearing diamonds, we modeled the change in the total He budget in a diamond due to radioactive production of $^4$He and removal by diffusion of $^4$He and $^3$He (details are in Methods—'Diffusion model of He in diamonds').

In order to understand the role of diffusion of He in diamonds, the important terms that control the contents of He are $r^2/D$ for both $^3$He and $^4$He (where $r$[L] is the diamond radius and $D$[L$^2$/T] is the diffusion coefficient) and $1/\lambda$ for $^4$He (where $\lambda$ [1/T] is the decay constants of U and Th isotopes). At $D$ values applicable to monocrystalline diamonds (10$^{-20}$–10$^{-21}$ cm$^2$ s$^{-1}$ [46]), $^4$He is almost immobile over hundreds of millions to billions of years (Fig. 3 and Fig. S1). In these cases, diffusion is primarily confined to the

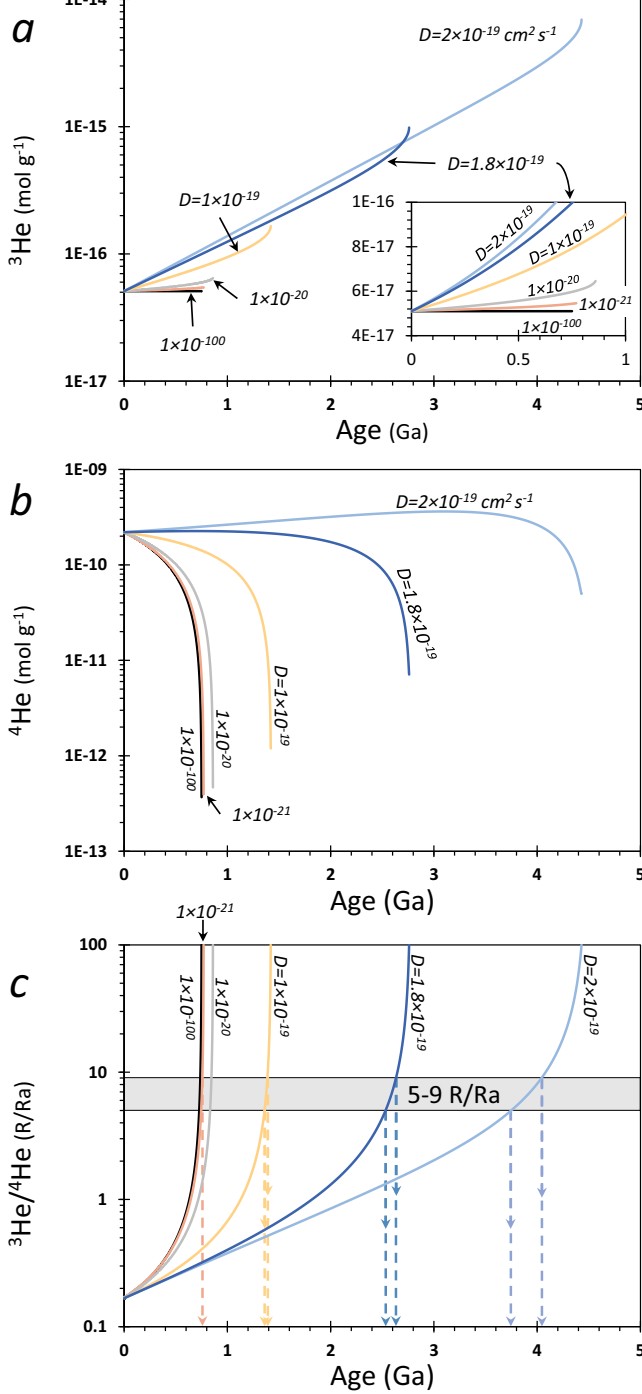

**Fig. 4 The effects of diffusion on model ages of HDF-bearing diamonds.**
**a–c** Back in time model calculations of $^3$He, $^4$He, and $^3$He/$^4$He, respectively, based on the present (measured) concentrations of U, Th, and He in a diamond (age = 0; the example shown is for diamond ON-FCH-349; model details are in "Methods—'Diffusion model of He in diamonds'"). If diffusion is important, it requires higher initial $^3$He (**a**) and $^4$He (**b**) concentrations, and more time, to reach the present-day $^3$He/$^4$He ratio (starting with the initial Ra ratio). The age of the diamond and the included HDF (i.e., $^3$He/$^4$He model age) is constrained for a given $D$ by the apparent asymptotic form of the curve (vertical arrows and dashed lines), assuming an initial $^3$He/$^4$He in the range of 5–9 Ra (**c**), which represent common values for MORB, the CLM and subducted components (Fig. 1). That is, the condition of "no diffusion" value of $D = 10^{-21}$ cm$^2$ s$^{-1}$ (yielding results essentially the same as $D = 10^{-100}$) only provides minimum model ages of diamonds, and depending on the diffusion rates, the true ages may be older.

permanently lost from the diamond by diffusion, while substantial $^4$He is lost by diffusion and added by radioactive decay. At $D \geq 1 \times 10^{-18}$ cm$^2$ s$^{-1}$, He loss by diffusion is fast, and the content of $^3$He and $^4$He in the diamond drops quickly after formation, having a major effect on the diamond's $^3$He/$^4$He ratio. Thus, within a portion of the range of available experimental diffusivities of $D = 10^{-16}$–$10^{-21}$ cm$^2$ s$^{-1}$ [44–47], our model results indicate that the budget of He in a diamond is affected by diffusion over geological time scales and its effects must be taken into account.

When considering $^3$He/$^4$He model ages of HDF-bearing diamonds (Fig. 4), the diffusion effects translate to older ages, which increases with increasing diffusivities and diamond age. This can be explained as follows: in the case of no diffusion, $^3$He remains constant and $^4$He is added over time due to production by radioactive decay of U and Th. With diffusion, $^4$He is added by radioactive decay, while $^3$He and $^4$He are lost by diffusion. As a result of the loss by diffusion, it requires more time, and higher initial $^3$He and $^4$He concentrations, to reach the present-day $^3$He/$^4$He ratio from any initial Ra value (Fig. 4); we assume an initial $^3$He/$^4$He in the range 5–9 Ra, representing common values for MORB, the CLM, and subducted components (Fig. 1). This means that the assumption of no diffusion (closed system, effectively for $D < 10^{-21}$) yields a minimum age of the diamond, while if diffusion is important, the true crystallization age is older, and depends on the diffusion history (Fig. 4c).

**The significance of microinclusion homogeneity.** Another important factor for determining the (U-Th)/He ages of natural HDF-bearing diamonds is the homogeneity of the microinclusion composition and density through the diamond. This is because U and Th compositions are obtained by laser ablation inductively coupled plasma mass spectrometry (ICP-MS), whereas the He isotope ratio and content are measured by bulk crushing and static MS analyses, two destructive methods that sample different diamond volumes, with the data then combined for (U-Th)/He age calculations.

As a rule, the major and trace-element composition of HDF microinclusions in an individual diamond is homogeneous, and to date, out of ~300 HDF-bearing diamonds, only a few show significant radial (core-to-rim) compositional changes, for example, refs. [18,34,60]. The microinclusion compositions are homogeneous in the De Beers Pool and Finsch diamonds analyzed here[49,50], as demonstrated by the close Th/U ratios of duplicate analyses of the same diamond (Fig. S2). Variations in trace-element concentrations in the diamond reflect variations in the spatial abundance of microinclusions. Such variations indicate that for some of the De Beers Pool and Finsch HDF-bearing

diamond's edge (i.e., for a diamond with $r = 2.5$ mm, there is <250 μm of diffusion in 1 Ga if $D < 10^{-21}$ and <750 μm of diffusion in 1 Ga if $D < 10^{-20}$) and thus it has little effect on the ingrowth of $^4$He and the evolving $^3$He/$^4$He ratio within the diamond. For example, the total loss of $^4$He and $^3$He after 500, 1000, and 3000 Ma is 4%, 5%, and 9%, and 5%, 7%, and 13%, respectively, and the change in $^3$He/$^4$He ratio is negligible (1.5%, 2.5%, and 4%, respectively). In such cases, (U-Th)/He approximates closed-system evolution. Even at $D = 1 \times 10^{-20}$ cm$^2$ s$^{-1}$ most of the diamond shows only small effects from diffusion over billions of years, with <37% of $^3$He and <27% of $^4$He lost over time periods of ~3000 Ma, which lowers the $^3$He/$^4$He ratio by 14%. At $D > 10^{-19}$ cm$^2$ s$^{-1}$, on the other hand, He loss is significant, with the effect that substantial $^3$He is

diamonds the inclusion-density scatter is small, whereas for others it is somewhat larger. The bulk trace-element composition of a diamond (including its U and Th content), however, can be closely represented by averaging a few laser ablation ICP-MS analyses of different parts in the diamond. This was shown by Rege et al.,[61] who compared the laser ablation averages of two HDF-bearing diamonds to their composition determined by INAA analyses[62] (Fig. S2), which average a much larger diamond volume, similar to He analyses[41,54].

The ability to approximate the bulk U and Th of an HDF-bearing diamond does not always allow for (U-Th)/He age determination. For example, in two carbonatitic HDF-bearing diamonds from Finsch, the internal microinclusion-bearing and overgrown microinclusion-free zones could not be entirely separated for He analyses. In these cases, a shortfall of He compared to U and Th (measured solely in the microinclusion-bearing zone) results in an apparent high (U-Th)/He ratios (Fig. 2b). In contrast, the $^3$He/$^4$He ratios of these diamonds are not affected by microinclusion-density variations. All other De Beers Pool and Finsch HDF-bearing diamonds analyzed in the present study had no such issues, and their (U-Th)/He are used for calculating ages (Fig. 2c and 5; excluding a single diamond mentioned above, with an unusually high U/Th ratio).

## Discussion

Helium isotope ratios of mantle samples provide fundamental geochemical information on their sources and can distinguish the origin of volatiles in the Earth interior[51,63,64]. For example, the measured $^3$He/$^4$He ratios of the South African saline HDF-bearing diamonds (2.7–5.2 Ra; Fig. 1a) and their "errorchron" initial value (4.6 ± 0.7 Ra; Fig. 2c) strengthen the connection between saline HDF and recycled subducted surface material[36], similar to "low-$^3$He/$^4$He" OIBs[52,63,65]. Extreme radiogenic ($^3$He/$^4$He « 1 Ra) signatures, on the other hand, cannot be directly related to any mantle component as sampled by oceanic or continental basalts[51,52,66] nor to surficial helium because helium is not recycled by subduction[67]. Therefore, the low $^3$He/$^4$He « 1 Ra in xenoliths from the Kaapvaal craton has been attributed to their Proterozoic emplacement age and post-emplacement production of radiogenic $^4$He in the host crystals (i.e., olivine, clinopyroxene), compared to xenoliths with Mesozoic emplacement ages in which $^3$He/$^4$He ≈ 5 Ra[68] (Fig. 1b and Supplementary Data 2[69], excluding three xenoliths of recycled oceanic-crustal protoliths from Roberts Victor). In diamonds, the low diffusivity of He (discussed below) allows radiogenic $^3$He/$^4$He signatures to develop from the time of diamond formation in the mantle, irrespective of their emplacement age by kimberlites. The South African silicic and carbonatitic HDF-bearing diamonds are such examples; they display radiogenic $^3$He/$^4$He ratios and correlated nitrogen aggregation characteristics (Fig. 2), which support their formation during earlier metasomatic events than those recorded by the saline HDF-bearing diamonds from the same kimberlites. A key issue, however, is whether the He retention characteristics in HDF-bearing diamonds allow meaningful geochronology by (U-Th)/He age determination, which can be used to unravel the timing of these metasomatic events.

The diffusion calculations have important implications for constraining the ages implied by the diamonds. While diffusivities lower than $D \sim 10^{-19}$ cm$^2$ s$^{-1}$ have only small impacts on model ages, a major difference occurs over small changes above that value (Fig. 4c). Additional constraints on the diffusion rates in HDF-bearing diamonds and therefore their ages can be achieved by considering independent geological evidence (Fig. 5). For example, diamond ON-DBP-332 yields an age of 0.4 Ga at $D = 1 \times 10^{-19}$ cm$^2$ s$^{-1}$ that increases to 1.2 Ga at $D = 4 \times 10^{-19}$ cm$^2$ s$^{-1}$, whereas

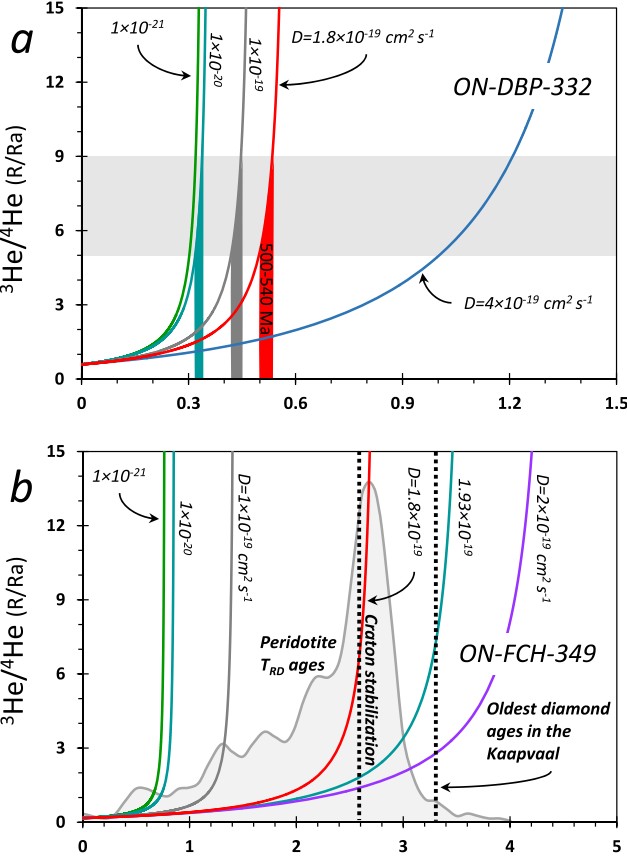

**Fig. 5 Diffusion impact and constraints on diamond $^3$He/$^4$He model ages.** $^3$He/$^4$He as a function of age and He diffusivity for **a** ON-DBP-332 (red diamond symbol in Fig. 2b) and **b** ON-FCH-349 (yellow circle in Fig. 2b), calculated based on the measured $^3$He/$^4$He and the $^4$He, $^3$He, U, and Th concentrations in these diamonds, and assuming HDFs initial Ra values between 5 and 9 (gray horizontal shading in (**a**); representing common values for MORB, the CLM and subducted components). The $^3$He/$^4$He model ages increase with increasing diffusivity. For ON-DBP-332 (**a**) the ages are between 320–340, 420–460, and 500–540 Ma (turquoise, gray, and red vertical shading, respectively). The potential range $^3$He/$^4$He model ages for diamond ON-FCH-349 (**b**) can be constrained by considering the oldest diamond ages in the Kaapvaal craton[19,71] and the distribution of Kaapvaal peridotite Re-depletion ages[70] ($T_{RD}$ model age; relative probability of $n = 228$; gray shading in **b**). As HDF-bearing diamonds are unlikely to survive the thermal and tectonic history prior to craton stabilization at ~2.6 Ga[72-74], this translates to a maximum diffusivity of He in HDF-bearing diamonds of $<1.8 \times 10^{-19}$ cm$^2$ s$^{-1}$. Thus, the timing of diamond formation and CLM metasomatism based on (U-Th)/He dating of HDF-bearing diamonds is constrained to be between $^3$He/$^4$He model ages for $D = 1 \times 10^{-21}$ and $1.8 \times 10^{-19}$ cm$^2$ s$^{-1}$. Additional details are given in the text.

ON-FCH-349 yields an age of 1.4 Ga at $D = 1 \times 10^{-19}$ cm$^2$ s$^{-1}$ that increases to 4 Ga at $D = 2 \times 10^{-19}$ cm$^2$ s$^{-1}$. The latter model age result is unreasonably old based on the known geological history of the Kaapvaal craton, as indicated by Re-depletion ages ($T_{RD}$) of peridotite xenoliths[70] (Fig. 5b). Moreover, when considering the oldest documented ages for monocrystalline diamonds from the Kaapvaal (~3.3 Ga)[19,71], the range of possible diffusivities is limited to $D \leq 1.93 \times 10^{-19}$ cm$^2$ s$^{-1}$ (for $^3$He/$^4$He$_0 \approx 8$ Ra). Compared to monocrystalline diamonds, HDF-bearing diamonds are less stable for long periods of time under mantle conditions and more likely to be resorbed. Such diamonds are unlikely to survive the thermal and

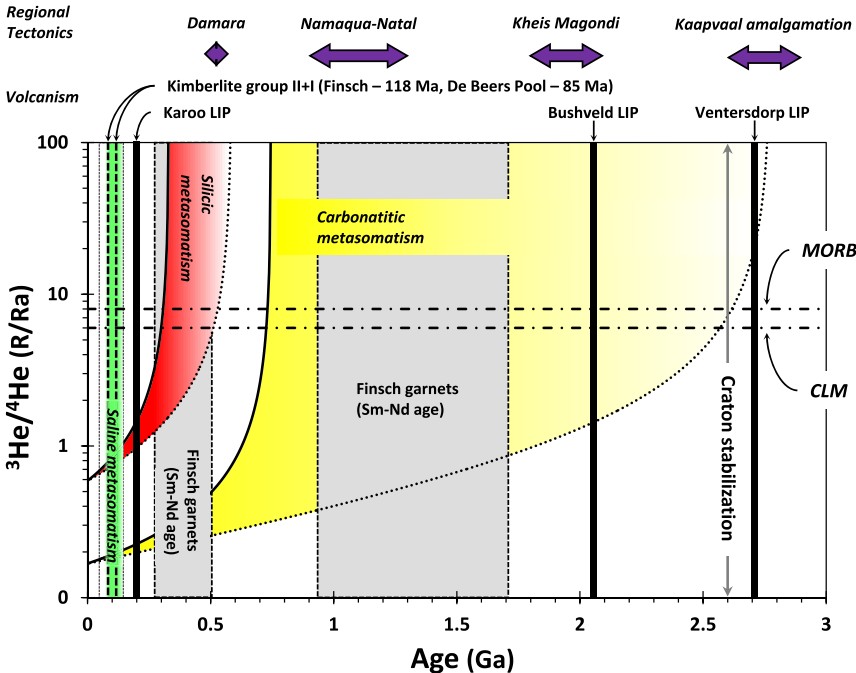

**Fig. 6 Carbon- and water-rich (C-O-H) metasomatic events recorded by HDF-bearing diamonds in the southwest Kaapvaal cratonic lithosphere.** U-Th-He geochronology of HDF-bearing diamonds from De Beers Pool and Finsch kimberlites reveal at least three episodes of different invading C-O-H fluid types affected the Kaapvaal continental lithospheric mantle (CLM) during the last ~2.6 Gyr. Saline fluids (green) controlled the metasomatism before/during late-Mesozoic kimberlite eruptions. An earlier silicic metasomatism (red) took place during the Paleozoic between 300–540 Myr; a time frame that coincides with CLM enrichment events at 391 ± 120 Ma as indicated by subcalcic garnets from Finsch[74], and overlaps the timing of the Damara orogeny (ca. 500 Ma)[76]. The oldest episode by carbonatite fluids (yellow) could take place throughout most of the Proterozoic, following craton stabilization at 2600 Ma and until 750 Ma. This event can be related to a mantle enrichment recorded in Finsch garnets (1720–940 Myr)[74] and the timing of the Namaqua–Natal Orogeny between 900 and 1250 Myr[75], but a connection to older mantle metasomatic events and large surface deformational events are also possible.

tectonic history prior to craton stabilization at ~2.6 Ga[72–74]. This translates to a maximum potential $^3$He/$^4$He model age of 2.6 Ga and thus an upper limit of $D \approx 1.8 \times 10^{-19}$ cm$^2$ s$^{-1}$, which we propose here as the best current estimate for the maximum diffusivity of He in HDF-bearing diamonds, until further constrained by experimental studies. The uncertainty on the average U content impacts this estimation by $\pm 0.2 \times 10^{-19}$ cm$^2$ s$^{-1}$ (i.e., $D_{\text{He in diamond}} = 1.8 \pm 0.2 \times 10^{-19}$ cm$^2$ s$^{-1}$). Therefore, diamond ON-FCH-349 formed from carbonatitic HDFs between 750 and 2600 Myr (Figs. 5 and 6) and based on the upper limit of $D \approx 1.8 \times 10^{-19}$ cm$^2$ s$^{-1}$, the timing of diamond formation of ON-DBP-332 during silicic metasomatism can be constrained between a minimum age of 300 Ma and a maximum of 540 Ma.

The Kaapvaal CLM in South Africa is one of the most studied Archean provinces and is commonly used as an exemplar for understanding the evolution of cratonic CLM[11,70,73]. Our new data on HDF-bearing diamonds reveal three episodes of chemical enrichment events in the southwest Kaapvaal CLM, each by a different C-O-H metasomatic agent, following the intensive chemical depletion by melting during its accretion in the Archean. The oldest episode by carbonatite fluids could have taken place sometime in the Proterozoic, after craton stabilization at 2600 Ma and until 750 Ma (Fig. 6). Within this large time frame, this event could possibly be tied with mantle enrichment in the southwest Kaapvaal CLM between 1720 and 940 Myr, as evidenced by subcalcic garnets with sinusoidal rare earth element patterns from Finsch[74], or the Namaqua–Natal Orogeny between 900 and 1250 Myr[75]. However, a connection to older mantle metasomatic events, which were likely coeval to large surface

tectonics and volcanism in the Kaapvaal craton, could be inferred as well (Fig. 6).

The time frame for the metasomatic event by Paleozoic C-O-H silicic fluids is more limited, between 300 and 540 Myr (Fig. 6). This time interval coincides with a second garnet enrichment event in Finsch between 300 and 500 Myr and overlaps with the Damara orogeny (ca. 500 Ma)[76], but is distinct from younger and older records of silicic metasomatism associated with the Karoo and Bushveld volcanism[77,78]. The youngest episode, by highly saline fluids, controlled the metasomatism that also led to diamond formation in the Kaapvaal CLM during the Cretaceous, as indicated by the majority of saline compositions in HDF-bearing diamonds with low-aggregated nitrogen (25 of 26) from De Beers Pool, Koffiefontein, and Finsch kimberlites[49,50,54,79], and their 90 ± 42 Ma age indicated by the positive trend on the (U-Th)/He isochron diagram (Fig. 2c). This metasomatism could reflect a single event before the Finsch eruption at 118 Ma, or a few episodes of saline fluid infiltration events that started before 118 Ma and continued until the De Beers Pool kimberlites eruptions at 85 ± 5 Ma (Fig. 6). Considering the incompatible chemical behavior of helium[80] and the susceptibility of the CLM to metasomatic enrichment processes, we propose that the similarity between $^3$He/$^4$He values of saline HDFs (2.7–5.2 Ra; Fig. 1b) and xenoliths with late-Mesozoic emplacement ages (5.1 ± 2.4 Ra) reflects the impact of the last metasomatic episode on the $^3$He/$^4$He isotopic composition of the CLM. This probably erased the record of previous C-O-H fluid episodes unless encapsulated in diamonds. Low $^3$He/$^4$He ratios are also documented for the De Beers Pool Wesselton kimberlite (1.6–3.7 Ra; Fig. 1b)[81], suggesting a possible link between saline HDFs and kimberlite through recycled

subducted surface material[36,81]. Further studies are needed to clarify possible genetic implications of this relationship, but it is emphasized here that the coeval timing of saline HDFs with late-Mesozoic kimberlite eruptions in the southwest Kaapvaal Craton strongly suggest such a link.

HDF-bearing diamonds are documented in many localities from all Archean cratons. Our diffusion model and constraints on He diffusivity in diamonds indicates an upper limit of $D \approx 1.8 \times 10^{-19}$ cm$^2$ s$^{-1}$, which provides a means to directly place an upper and lower limit on the timing of diamond crystallization and C-O-H metasomatism in different CLM provinces, by U-Th-He geochronology of HDF-bearing diamonds. For the southwest Kaapvaal CLM, the new findings thus delineate a sequence of different types of invading diamond-forming C-O-H fluid types over its history, carbonatitic during the Proterozoic, silicic during the Paleozoic, and saline during the Cretaceous, that led to fertilization, oxidation, and hydration of its deepest parts through space and time. The apparent link between metasomatism, HDF-bearing diamond formation, and tectonics and volcanism at Kaapvaal craton (Fig. 6), elucidates possible relationships between events occurring on Earth's surface and in the shallow CC and chemical events involving different C-O-H fluid types in the mantle below.

## Methods

### Samples, preparation, and analytical methods.
HDF-bearing diamonds from the De Beers Pool and Finsch kimberlites were selected from two suites that were previously analyzed for their nitrogen content and aggregation, and HDF micro-inclusions major and trace-element compositions[49,50].

Helium isotope analyses were performed at the Noble Gas Lab at the Lamont–Doherty Earth Observatory of Columbia University. After coarse crushing of the diamond, inner microinclusion-bearing fragments were selected and cleaned ultrasonically in a mixture of concentrated HF and concentrated HNO$_3$ for >2 h, washed with ethanol and milli-Q water, dried at 130 °C, weighed, and loaded in a stainless-steel electromagnetic crusher. Gas extraction was performed under vacuum by sequential crushing with an automated piston (180-piston cycles in each crushing sequence) to obtain the He gas preferentially trapped in the microinclusions[58,82]. The amount of gas released over time from the different crushing sequences follows a power law until the diamond is no longer releasing He (>95% extraction); total helium contents and isotopic composition for each diamond are reported in Supplementary Data 1[48].

The extracted gas was purified by passage through a liquid nitrogen cooled charcoal trap at 77 K and by exposing the gas to a SAES getter at room temperature, before it was collected at 14 K on a chryogenically cooled trap. Helium was released from the charcoal at 45 K and abundance and isotopic analysis was performed with a MAP 215-50 (Mass Analyzer Products, Manchester, UK) noble gas mass spectrometer by peak jumping[83,84]. $^4$He is detected with a Faraday cup, and $^3$He with a channeltron multiplier operated in ion counting mode. The mass spectrometric system is calibrated with known aliquots of helium from gas collected in Yellowstone National Park (the MM Standard with $^3$He/$^4$He = 16.45 Ra[85,86]). Standard reproducibility during the period of the measurements was ~1% (1σ) and ~1.5% (1σ) for $^4$He and $^3$He, respectively. At room temperature, the crusher plus mass spectrometer blank is typically $1 \times 10^{-10}$ ccSTP (cubic-centimeters-at-standard-temperature-and-pressure) for $^4$He, and its $^3$He/$^4$He ratio is indistinguishable from atmospheric ratio within error[84]. The reported uncertainties include standard reproducibility, internal measurement precision, uncertainty associated with the nonlinearity correction, and the uncertainty in the blank subtraction.

### U-Th-He isochron ages.
An isochron approach is used for saline HDF inclusion-rich diamonds, based on a simplification of the general equation for closed-system evolution of $^4$He/$^3$He ratios[87]. Given that the helium in a sample under closed-system conditions reflects the initial amount plus the amount produced by radioactive decay,

$$^4\text{He}/^3\text{He} = {}^4\text{He}/^3\text{He}_i + {}^4\text{He*}/3\text{He} \qquad (1)$$

where $^4$He/$^3$He$_i$ is the initial ratio and $^4$He* is the amount of radiogenic $^4$He produced by the decay of U and Th. The amount of radiogenic $^4$He* produced over time in turn is equal to

$$^4\text{He*} = 8 \cdot {}^{238}\text{U} \cdot (e^{\lambda 238 t} \cdot -1) + 7 \cdot {}^{235}\text{U} \cdot (e^{\lambda 235 t} \cdot -1) + 6 \cdot {}^{232}\text{Th} \cdot (e^{\lambda 232 t} \cdot -1), \qquad (2)$$

where $t$ is the age of formation and $^{238}$U, $^{235}$U, and $^{232}$Th are the molar concentrations. Using the approximation $e^x \approx (1 + x)$, Eq. (2) can be simplified to:

$$^4\text{He*} = t \cdot \{8 \cdot \lambda_{238} \cdot {}^{238}\text{U} + 7 \cdot \lambda_{235} \cdot {}^{235}\text{U} + 6 \cdot \lambda_{232} \cdot {}^{232}\text{Th}\} = t \cdot (\text{U} + \text{Th})^* \quad (3)$$

where (U + Th)* is the present-day production rate of $^4$He. Therefore, Eq. (1) can be expressed as

$$^4\text{He}/^3\text{He} = {}^4\text{He}/^3\text{He}_i + t \cdot \{8 \cdot \lambda_{238} \cdot {}^{238}\text{U} + 7 \cdot \lambda_{235} \cdot {}^{235}\text{U} + 6 \cdot \lambda_{232} \cdot {}^{232}\text{Th}\}/^3\text{He} \qquad (4)$$

and

$$^4\text{He}/^3\text{He} = {}^4\text{He}/^3\text{He}_i + t \cdot (\text{U} + \text{Th})^*/^3\text{He} \qquad (5)$$

(U + Th)* can be further simplified to (U + Th)* = (1.291. $^{238}$U + 0.2969. $^{232}$Th) Ga$^{-1}$, $^{238}$U/$^{235}$U = 137.88, $\lambda_{238}$ = 0.155125 Ga$^{-1}$, $\lambda_{235}$ = 0.98485 Ga$^{-1}$, and $\lambda_{232}$ = 0.04948 Ga$^{-1}$.

Equation (5) shows that under closed-system conditions, a series of samples formed at the same time with the same initial $^4$He/$^3$He will fall on a line on an "isochron" plot of $^4$He/$^3$He vs. (U + Th)*/$^3$He (Fig. 2c), and the slope will correspond to the age, as long as approximation $e^x \approx (1 + x)$ is valid. At 100 Ma, the approximate age for our saline HDF diamonds, the offset is only ~0.6% (600 kyr); thus, the approximation and the "isochron" approach works well for these samples. We note, however, that with older ages, the true age increasingly diverges from the production rate age estimate such that, for example, at 1.0 Ga the offset is 5.9% (59 Ma), at 2.0 Ga it is 11.6% (132 Ma), and at 3.0 Ga it is 17.1% (515 Ma). We calculate our ages and errors using IsoplotR[87], which argues that because the parent and daughter nuclides are analyzed separately, their uncertainties are uncorrelated and the isochron regression can be done using the simple least-squares fitting method of York[88].

### Diffusion model of He in diamonds.
We assume that during the metasomatic event, the crystallizing diamond and its microinclusions are in equilibrium with respect to He content. This assumption is valid based on the short time it takes to reach equilibrium after diamond formation. According to the large amount of inclusions, their micrometer size and the distance between neighboring inclusions (a few micrometers on average[33,89]), equilibrium between the microinclusions and the diamond will be reached shortly (<1 Ma) after formation by diffusion. This conclusion is based on the time scale of diffusion ($r^2/D < 0.3$ Ma) for a micro-inclusion size of ~1 μm and diffusion coefficient ($D$) of $10^{-21}$ cm$^2$ s$^{-1}$. Continuous diffusion only affects the He budget in the diamond and microinclusions if He actually diffuses out of the diamond. We also assume that He is lost forever (He = 0) once outside the HDF-bearing diamond (which explains the boundary conditions in Eq. (6), below). This is based on the fact that diffusion coefficients for mantle minerals surrounding the diamond are many orders of magnitude higher (e.g., Ol $10^{-3}$–$10^{-4}$ cm$^2$ s$^{-1}$; Opx $10^{-2}$ cm$^2$ s$^{-1}$; Cpx $10^{-2}$ cm$^2$ s$^{-1}$) and He contents are much lower than in HDF-bearing diamonds[90–95]. The change in the $^4$He budget in the diamond, therefore, equals the production of $^4$He by radioactive decay of U and Th in the microinclusions minus the removal of $^4$He from the diamond due to diffusion (assuming a constant diffusion coefficient) according to Fick's second law. We assume a sphere and radial symmetry in the distribution of He in the diamond, as well as homogeneous microinclusion composition and density, for example, refs. [33,96]. Moreover, in the absence of experimental constraints about possible differences between $^3$He and $^4$He diffusivities in diamonds, we assume that $D(^3\text{He}) \approx D(^4\text{He})$. The mass balance equation of He, as a function of the distance from the diamond's center ($r$) and time, taking into account both diffusion and radioactive decay is:

$$\frac{\partial C}{\partial t} = f(t) + \frac{1}{r^2}\frac{\partial}{\partial r}\left(r^2 D \frac{\partial C}{\partial r}\right) \qquad C(R, t) = 0 \qquad C(r, 0) = C_0 \qquad (6)$$

where $C$ [M/L$^3$] is the $^4$He content, $t$ [T] is the time since the diamond formed, $r$ [L] is the distance from the center of the diamond, $R$ [L] is the diamond radius, $D$ [L$^2$/T] is the diffusion coefficient and $C_0$ [M/L$^3$] is the initial concentration of $^4$He. $f(t)$ is the production rate of $^4$He from radioactive decay of $^{238}$U, $^{235}$U, and $^{232}$Th described as:

$$f(t) = {}^{238}\text{U}\{8 \cdot \lambda_{238} \cdot e^{\lambda_{238}(T-t)} + 7/137.88 \cdot \lambda_{235} \cdot e^{\lambda_{235}(T-t)} + 6 \cdot K \cdot \lambda_{232} \cdot e^{\lambda_{232}(T-t)}\} \qquad (7)$$

where $^{238}$U [M/L$^3$] is the $^{238}$U concentration today, $\lambda_{238, 235,}$ and $_{232}$ [1/T] are the decay constants of $^{238}$U, $^{235}$U, and $^{232}$Th respectively, $T$ [T] is the diamond formation age, $t$ is an age between $T$ and today, $^{238}$U/$^{235}$U is the atomic ratio of 137.88 today, and $K = {}^{232}$Th/$^{238}$U today.

The concentration of $^3$He with time is determined only by diffusion of helium according to Fick's second law (i.e., $f(t) = 0$ in Eq. (6)). Equation (6) does not have an analytical solution, and therefore we solved[97] it by finite difference using the Crank–Nicolson method[98] (the equation derivation is given below). To evaluate the change in the total He budget in the diamond as a function of time for different He diffusivities, between $D = 10^{-21}$ and $10^{-18}$ cm$^2$ s$^{-1}$, we solve Eq. (6) and integrate it across the diamond using a hypothetical example. Figure 3 and Fig. S1 shows forward modeling results for the change in $^4$He and $^3$He content and $^3$He/$^4$He ratio with time after diamond formation, in a diamond with $r = 2.5$ mm, $^3$He$_0$ = $1.1 \times 10^{-15}$ mole g$^{-1}$, and $^4$He$_0$ = $1 \times 10^{-10}$ mole g$^{-1}$ (i.e., 8 Ra), U = $5 \times 10^{-10}$

mole g$^{-1}$, and Th/U = 4, for different He diffusivities. These parameters are approximately average values of the studied De Beers Pool and Finsch diamonds and close to those of available HDF-bearing diamonds[54,59]. The model results indicate that within the range of available experimental He diffusivities, the budget of He in a diamond is affected by diffusion over geological time scales (further details are in the main text).

$^3$He/$^4$He model ages of HDF-bearing diamonds are calculated back in time, based on the present-day (measured) concentrations of U, Th, and $^4$He and $^3$He (e.g., Figure 4; an example is shown for diamond ON-FCH-349; Supplementary Data 1[48]). This is done by iterations and the best fit of the numerical solution of Eq. (6) to the measured concentrations of $^3$He and $^4$He. The model runs for a range of ages, in which for each age the $^3$He$_0$ is calculated according to the analytical solution of $^3$He vs. time:

$$^3\text{He}_0 = {}^3\text{He}_{(\text{measured})} \Big/ \left( \frac{6}{\pi^2} \sum_{n=1}^{\infty} \frac{1}{n^2} \exp\{-Dn^2\pi^2 t / R^2\} \right) \text{(Eq. 6.20 in ref.}^{98}\text{).} \quad (8)$$

Thus, for a measured $^3$He/$^4$He and U/Th/He today, the model determines the ($^3$He/$^4$He)$_0$ at any point in time under the determined diffusion conditions set by the value of $D$. The age of the diamond and the included HDF (i.e., the $^3$He/$^4$He model age) is constrained for a given $D$ by the apparent asymptotic form of the curve, assuming an initial ($^3$He/$^4$He)$_0$ in the range 5–9 Ra, which represents common values for MORB, the CLM, and subducted components (e.g., Fig. 1).

The effects of diffusion on $^3$He, $^4$He, and $^3$He/$^4$He through time are illustrated in Fig. 4. In each of the calculations, the model starts in the past, with an initial Ra value, and calculates how long it takes to reach the measured (present) concentrations of $^4$He and $^3$He. Since the formation of the diamond in the past, $^4$He is added by radioactive production of U and Th and at the same time, $^3$He and $^4$He are removed by diffusion. In the case of no diffusion, $^3$He is constant with time, and $^4$He is only added by production. As a result of the removal by diffusion, at any point in time the $^3$He/$^4$He will be lower with diffusion than with no diffusion (see also Fig. 3). Therefore, if diffusion is important it requires higher initial $^3$He and $^4$He concentrations and more time to reach the $^3$He/$^4$He present ratio (starting with the initial Ra ratio; Fig. 4). That is, the condition of "no diffusion" yields the minimum estimate of the diamond formation age, and depending on the diffusion rates, the true ages may be older (further details are in the main text).

Crank–Nicolson development for Eq. (6):

$$\frac{\partial C}{\partial t} = \frac{2D}{r}\frac{\partial C}{\partial r} + D\frac{\partial^2 C}{\partial r^2} + f(t)$$

$$\frac{\partial C}{\partial t} \Rightarrow \frac{C_i^{j+1} - C_i^j}{\Delta t}$$

$$\frac{\partial^2 C}{\partial r^2} \Rightarrow \frac{1}{2(\Delta r)^2}\left( \left( C_{i+1}^{j+1} - 2C_i^{j+1} + C_{i-1}^{j+1} \right) + \left( C_{i+1}^j - 2C_i^j + C_{i-1}^j \right) \right)$$

$$\frac{\partial C}{\partial r} \Rightarrow \frac{1}{2}\left( \frac{\left( C_{i+1}^{j+1} - C_{i-1}^{j+1} \right)}{2(\Delta r)} + \frac{\left( C_{i+1}^j - C_{i-1}^j \right)}{2(\Delta r)} \right)$$

$$C \Rightarrow \frac{1}{2}\left( C_i^{j+1} + C_i^j \right)$$

$$A1 = \frac{1}{2}\left( r_i - \frac{1}{\Delta r} \right)$$

$$A2 = \frac{\Delta r}{D\Delta t} + \frac{1}{\Delta r}$$

$$A3 = -\frac{1}{2}\left( r_i + \frac{1}{\Delta r} \right)$$

$$B1 = -A1$$

$$B2 = \frac{\Delta r}{D\Delta t} - \frac{1}{\Delta r}$$

$$B3 = -A3$$

$$AA = \begin{bmatrix} A1+A2 & A3 & 0 & \dots & \dots & \dots & 0 \\ 0 & A1 & A2 & A3 & 0 & \dots & 0 \\ 0 & 0 & A1 & A2 & A3 & \dots & 0 \\ \vdots & & \ddots & \ddots & \ddots & \ddots & \vdots \\ \vdots & & \ddots & \ddots & A1 & A2 & A3 & 0 \\ 0 & \dots & & \ddots & 0 & A1 & A2 & A3 \\ 0 & \dots & \dots & 0 & 0 & A1 & A2 \end{bmatrix}$$

$$BB = \begin{bmatrix} B1+B2 & B3 & 0 & \dots & \dots & \dots & 0 \\ 0 & B1 & B2 & B3 & 0 & \dots & 0 \\ 0 & 0 & B1 & B2 & B3 & \dots & 0 \\ \vdots & & \ddots & \ddots & \ddots & \ddots & \vdots \\ \vdots & & \ddots & \ddots & B1 & B2 & B3 & 0 \\ 0 & \dots & & \ddots & 0 & B1 & B2 & B3 \\ 0 & \dots & \dots & 0 & 0 & B1 & B2 \end{bmatrix}$$

$$C_N = \begin{bmatrix} 0 \\ C_0 \\ \vdots \\ C_0 \end{bmatrix}$$

$$\left[ C^{j+1} \right] = \left[ AA^{-1} \right]\left( [BB]C^j + f^{j+1} + [C_N] \right)$$

$$C_{\text{tot}}^{j+1} = \frac{3\sum_{i=1}^{n-1} C_i^j r_i^2}{R^3 \Delta r} + \frac{3C_n \Delta r}{2R}$$

where $j$ is the time index, $i$ the spatial index, $n$ the number of time steps, $C_{\text{tot}}$ the integrated concentration in the diamond, **f** is a vector with the number of cells according to the spatial discretization, with the production term of He ($f(t)$), and $R$ the diamond radius.

## Data availability

Sample metadata have been archived in the System for Earth Sample Registration (SESAR) with associated International GeoSample Numbers (IGSNs). The new data (Supplementary Data 1, https://doi.org/10.26022/IEDA/111776) as well as the compiled xenolith data set used here (Supplementary Data 2, https://doi.org/10.26022/IEDA/111777) have been submitted to EarthChem (www.earthchem.org/petdb), and are provided as Supplementary data sets linked to this paper.

## Code availability

The new code based on our model in "Methods—'Diffusion model of He in diamonds'" have been submitted to Zenodo (Diamonds diffusion model used in Weiss et al., 2021, *Nature Communications*) and is available at https://zenodo.org/record/4329753#.X9qCjrOxU2w.

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

## Acknowledgements

Y.W. acknowledges ISF Grant No. 2015/18; Y.W., S.L.G., C.C., G.W., and Y.K. acknowledge support by NSF Grants EAR-1348045 and EAR-1725323. S.L.G. acknowledges support from the Storke Endowment of the Department of Earth and Environmental Sciences of Columbia University. We thank O. Navon for constructive discussions, L.D. Baker and R. Friedrich for their help with the He analyses, and A. Kiro, B. Oryan, and G. Weiss for discussions and help with formatting the figures. This is Lamont–Doherty Earth Observatory contribution number 8493.

## Author contributions

Y.W., S.L.G., and C.C. conceived and developed the project. Y.W. performed the He analyses with G.W. at LDEO. Y.K. constructed and solved the diffusion (mass balance) model of He in diamonds. Y.W., S.L.G., C.C., Y.K., and G.W. wrote the paper. J.W.H. provided the diamond samples for the study and contributed intellectually to the paper.

## Competing interests

The authors declare no competing interests. Readers are welcome to comment on the online version of the paper.
