## [Peer Review File · Nature Communications]

REVIEWER COMMENTS

Reviewer #1 (Remarks to the Author):

This study reports new He isotope data for a suite of diamonds rich in high-density fluid (HDF) inclusions, sourced from two Cretaceous localities in the West Kaapvaal, South Africa. These diamonds had been previously studied for their fluid inclusion compositions, including U and Th concentrations, and nitrogen aggregation state. New He isotope data are combined with previously published results and a novel He diffusion model to establish the crystallisation age of these diamonds by using the (U+Th)/He dating method. These estimated crystallisation ages are then compared with those of major tectonic and magmatic events in southern Africa, including metasomatic events recorded in the radiogenic isotope systematics of mantle xenoliths, to establish a sequence of diamond-forming metasomatic events in the West Kaapvaal.

The objectives of the study are clear, the quality of the data is good and the approach to obtain diamond crystallisation ages is as robust as it can be given the complexities and limitation of this new dating method (new to date diamonds). I especially praise the Authors for developing a novel method to calculate (U+Th)/He ages which keeps into account He diffusion out of diamond, something not considered in the only previous published effort to get diamond ages by the (U+Th)/He method (Timmermann et al., 2019 cited).

Overall, I am supportive of publication of this work in Nature Communications because it provides a new approach to directly date diamond formation ages. Yet I urge the Authors to carefully consider my comments below. Some are mainly directed to improve the impact and clarity of this work, others are essentially "words of caution" because some of the conclusions are not fully supported by the results or sounds quite speculative.

1) The introductory section, while well framed, overlooks an important topic, that is the well-established application of radiogenic isotope dating of silicate and sulfide inclusions in diamonds to obtain diamond formation ages, including the current debate regarding the meaning of these mineral inclusion ages. In this sense, direct dating of HDF-rich diamonds can provide a means to circumvent this debate and get diamond crystallisation ages with potentially fewer assumptions than for mineral inclusion dating. Hence, I believe the scope and impact of this work could be made broader by including this aspect of diamond dating studies rather than focussing exclusively on the importance of dating HDF-rich diamonds.

2) Pooling of diamond data for the Kimberley (De Beers pool) and Finsch mines is risky. Kimberley and Finsch are ~100 km apart and they sample a very different mantle column based on existing mantle xenolith and xenocryst data (e.g., Griffin et al., 2003 Lithos, cited). Their different emplacement ages underpin a significant modification of the lithospheric mantle underneath this region between 118 and 85 Ma. In other words, the De Beers diamonds could reflect a different event than the Finsch diamonds, and this should be recognised in the revised manuscript.

3) The maximum diffusion coefficient DHe calculated by the Authors is based on the assumption that the oldest diamond-forming event in the Kaapvaal was coeval (or postdated) the final assembly of this craton at ~2.6 Ga (L143 and thereabouts). Yet, there are diamonds with 'kimberlitic' features in the Witwatersrand conglomerates which must be older than the conglomerate deposition age of ~2.9 Ga. I would rather use an upper bound of 3.2-3.3 Ga (see Smart et al., 2016 Nature Geo) to calculate maximum DHe.

4) I am sorry to say that I find the correlation between the Proterozoic diamond forming event and the Namaqua-Natal orogeny highly speculative (L155-160 and thereabouts). There are multiple events occurring in and around the Kaapvaal between >2.6 Ga and 750 Ma (i.e. the permissible ages of diamond formation based on He isotope systematics) beyond the Namaqua-Natal orogeny, including the Bushveld LIP and the Kibaran orogeny just to mention a couple. So, why should this diamond age specifically reflect the Namaqua-Natal orogeny? Are there other diamond formation ages based on mineral inclusions at Finsch which could provide additional support to a link between diamond formation and the N-N orogeny? The Authors employ Ar-Ar ages of mantle-derived phlogopite to support the conclusion of mantle metasomatism underneath the West

Kaapvaal during the N-N orogeny. However, Ar/Ar dating of mantle phlogopite is known to provide spurious ages (Phillips and Onstott, 1988 *Geology*; Phillips, 1991 *Chem Geol*; Giuliani et al., 2012 *IKC Ext Abstr*).

5) More broadly, the West Kaapvaal has experienced an endless sequence of tectonic and magmatic events, which may (or may not) have left permanent metasomatic 'scars' on the underlying lithospheric mantle. While the Authors clearly demonstrate a link between different diamond-forming events, mantle fluid compositions and potentially ages, the sequence of mantle fluid infiltration they present sounds too 'definitive'. For example, there is ample evidence of silicate melt metasomatism associated with Bushveld and Karoo (Hoal, 2003 *Lithos*; Giuliani et al., 2014 *EPSL*) yet they did not observe a link between formation of silicic HDF-bearing diamond and these major events (maybe because these were diamond-destructive events or simply because of the limited size of the dataset). In simple words, diamonds alone cannot constrain the metasomatic history of the Kaapvaal and a more integrated approach should also include mantle xenoliths and magmatic events. In other words, I would tone down the last part of the discussion and focus the conclusions of this work on the new methodology here developed and its applicability to diamonds elsewhere.

6) L172-175. This temporal relationship between formation of diamonds containing saline HDF and Mesozoic kimberlite (plus lamproite) magmatism is interesting. However, I shall note that kimberlite metasomatism of the lithospheric mantle in the western Kaapvaal produces carbonate-rich mantle xenoliths (e.g., Dawson et al., 2001 *CMP*; Soltys et al., 2016 *Lithos*; Fitzpayne et al., 2018 *Mineral Petrol*) with only limited evidence of addition of Cl and Na (e.g., Giuliani et al., 2012 *Geology*; Giuliani et al., 2013 *CMP*). So, while the temporal correlation holds, the genetic implications of this correlation deserve further work, which could also represent a further research avenue stemming out of this important study.

7) L176-180. I must strongly disagree with this statement and in fact with the article of Yaxley et al. which the Authors refer to here. Kimberlites are sub-lithospheric melts (Smith, 1983 *Nature*; Torsvik et al., 2010 *Nature*; Stagno et al., 2013 *Nature*; Stamm and Schmidt, 2018 *EPSL*; Pearson et al., 2019 *Elements*; Giuliani et al., 2020 *Sci Adv*; etc etc) and their only connection with the cratonic lithosphere is due to the deep melt extraction depths in the asthenosphere provided by thick cratonic roots. Kimberlites do not derive from previously metasomatised SCLM as argued originally by Dan McKenzie (Tainton and McKenzie, 1994 *J Pet*) based on their extrapolation of a petrological model that is only valid for basalts and similar melts, not kimberlites. This is a fundamental mistake that has been perpetuated for decades and which has only recently been finally clarified. The isotope geochemistry of kimberlites and link to mantle plumes require a sub-lithospheric source for kimberlites (see additional arguments in the references above).

8) It should be clearly explained how (U+Th)/He ratios were calculated because these element concentrations were measured in very different ways (LA-ICPMS vs bulk crushing + static MS). If I recall correctly one of the early works of Weiss et al. positively compared trace element concentrations of HDF-rich diamond obtained by LA-ICPMS and bulk-combustion (or a similar bulk method). I recommend reporting these pieces of information and demonstrate that U and Th concentrations by LA-ICPMS are representative of the bulk of fluid inclusions. This will make the methodology more sound to the reader.

Minor comments:

L16-17: This statement might be a bit too strong. At the end of the day, dating of mineral inclusions in diamonds essentially provides ages of diamond-producing metasomatic events in the lithospheric mantle. The novelty of this work is rather that (U-Th)/He dating may be a direct rather than indirect means to date diamond crystallisation events.

L24-25. I would rather say that the ages of these diamond-forming events are coveal with those of major tectonic events recorded in the crust. Also note that the link between mantle metasomatism and tectonic events in previous studies of mantle xenoliths is based on model ages (Lazarov et al.) and Ar/Ar dating (Hopp et al., both cited) where Ar/Ar dating of mantle phlogopite is well known to

provide spurious ages (Phillips and Onstott, 1988 Geology; Phillips, 1991 Chem Geol). So these links are not very robust.

L34. Some of these refs are probably not ideal in this context (e.g., Carlson and Irvine 1994; Hawksworth et al 1990; Mysen 1983). There are much more recent references with a strong focus on the composition of mantle fluids, both from my group (e.g., Giuliani et al., 2012 Geology; Giuliani et al., 2015 Nature Comms) and others (e.g., see recent papers in Lithos by Jurgen Konzett; the various papers by Maria-Luce Frezzotti on mantle fluids, e.g., 2012 EPSL; 2010 GCA; and the 2001 review of Anderson and Neumann in Lithos).

L75. This $3\text{He}/4\text{He}$ range is very similar to that of magmatic fluids in olivine from the Wesselton kimberlite (Giuliani et al., 2020 EGU abstract; attached), which is part of the Kimberley kimberlites where the De Beers Pool diamonds were sourced.

L93. Please include references for these emplacement ages.

L98. I shall also note that southern African Cretaceous kimberlites (Kimberley, but not Finsch, which is an olivine lamproite) feature a strong HIMU component (e.g., Smith, 1983 Nature; Collerson et al., 2010 Phys Chem Earth). HIMU OIBs (Hanyu and Kaneoka, 1997 Nature) and the HIMU-like Wesselton kimberlite (Giuliani EGU abstract) both contain low $3\text{He}/4\text{He}$ similar to these diamonds, which might further support the occurrence of recycled crustal material in the mantle source of these diamond-forming fluids.

L123. based on diamond of which size?

L166. I would not use the term "control" here given the abundant evidence of metasomatism by carbonate-rich and silicate melts in the Kaapvaal during the Mesozoic recorded in mantle xenolith studies (some examples in articles from my group just to give an idea)

Figure 1. I guess the Authors only compiled representative He isotope data for mantle xenoliths given that there are many more studies available than those in the "data compilation" file (e.g., Barry et al., 2015 Lithos; Dunai and Baur, 1995 GCA; Patterson et al., 1994 GCA; Porcelli et al., 1986 GCA and 1992 CMP; Yamamoto et al., 2004 Chem Geol).

Figure 1. My EGU2020 abstract (n. 5267) also reports a $3\text{He}/4\text{He}$ range of 1.6-3.7 R/Ra for fluid-inclusion rich olivine grains for the Wesselton kimberlite. It could perhaps be helpful to include these data as representative of the He isotope composition of the kimberlite host (at least for the De Beers Pool diamonds).

L462-463. This method (i.e. crushing of inner diamond fragments) avoids potential issues with alpha-implantation and ejection. I would stress this somewhere in the main text.

Additional minor suggestions and stylistic comments are included in the attached pdf.

Overall this is a nice and important contribution which advances the toolkit available to understand mantle processes. Hence, I reiterate my support towards its publication.

I hope the Authors will find my comments helpful and I am available for any clarification.

Andrea Giuliani

Zurich, 12-Aug-2020

Reviewer #2 (Remarks to the Author):

Diamonds provide unique opportunities to explore the history of processes occurring in the mantle. Dating diamonds, particularly zoned crystals or deposits with multiple generations of diamond formation is one of the most important challenges in the field. Dating fluid inclusions with (U-Th)/He methods is therefore very exciting, particularly if it can be related to different fluid compositions and defect concentrations in host diamonds. The study has the potential to be of sufficient interest to be considered for Nature Communications, but the quality of the execution and the significance of the findings needs to match the potential of the method.

Overall recommendation – I do not recommend that the paper is published in Nature Communications in its current form for the following reasons.

1. Although the concept is simple (and good) the detail is extremely important for this paper. I found it very hard to follow the detail in a paper of this style, and the logic of the story was not always clear to me. I wasted a lot of time poring over the supplementary information, extended data and previous papers by the authors. I think the paper would be much better suited a longer paper format where the methods, materials and logical flow of the arguments would be clearer.
2. I am not convinced by the novelty compared with previous papers. The main new contribution is the diffusion modelling, which is the aspect I am least convinced by.
3. I found the description of the analysis and diffusion modelling to be confusing. At one point it is stated that He was analysed in "inner" microinclusion-bearing portions of the sample. As far as I could understand only gas from the microinclusions was analysed, not He from the diamond lattice. There is also a discussion of where inclusions are in the samples, but no images. The modelling appears only to apply to a homogeneous distribution of inclusions (if I understand it correctly) but images in Weiss et al 2018 and Weiss and Goldstein 2018 of the same samples, clearly show variable core-rim relationships. Thus the distance between inclusion-rich zone and diamond surface would be very important (rather than the overall radius of the diamond). On top of these issues, the diffusivity of He (and the assumed equivalence of ^3He and ^4He) are so poorly known I would question the value of the modelling results. Finally, there is a relationship between measurable quantities (N concentration and aggregation), time (the output of the paper) and temperature (a major controlling factor in He diffusivity), so an opportunity for a more sophisticated iterative approach has been missed.

Other issues.

1. No uncertainties are given in extended data table 1, and no uncertainties are propagated or discussed. There are no error bars in fig 1a, 1b or 2a, but there are in 2b and 2c. The best fit in 2c is not an isochron, as only 3/5 (3/6 depending on selection) lie on the line within the stated errors. It is an errorchron. This must be explained in any reworking of the paper. Either the errors are much larger than stated, or the broad "isochron-like trend" arises from process other than radioactive decay.
2. The authors are well aware that N aggregation is affected by all of time, temperature and total N concentration, so plotting $^4\text{He}/^3\text{He}$ vs N aggregation is a strange choice that is never justified in the text.
3. In Extended data figure 1, why does the function for 50 Ma look so different from all the other times? It looks very strange. Are you sure it is not an error?

In summary, the paper has the potential to be an exciting and valuable contribution to the field. However, there are a lot of potential problems with the data and the modelling, so the paper needs to be written in a way that the reader can follow more exactly what has been done so that the uncertainties and limitations are transparent. The short paper format is not suitable for the depth of treatment that is required.

Response to reviewers: Helium in diamonds unravels over a billion years of craton metasomatism, by Yaakov Weiss, Yael Kiro, Cornelia Class, Gisela Winckler, Jeff W. Harris, and Steven L. Goldstein.

We thank the two reviewers for their thorough and thought-provoking reviews. We believe that we have addressed their concerns and that this made the paper stronger, and we hope they agree that it is now acceptable for publication.

The reviewers comments are italicized below and our responses are integrated within the text in blue non-italic text.

REVIEWER COMMENTS

Reviewer #1 (Remarks to the Author):

This study reports new He isotope data for a suite of diamonds rich in high-density fluid (HDF) inclusions, sourced from two Cretaceous localities in the West Kaapvaal, South Africa. These diamonds had been previously studied for their fluid inclusion compositions, including U and Th concentrations, and nitrogen aggregation state. New He isotope data are combined with previously published results and a novel He diffusion model to establish the crystallisation age of these diamonds by using the (U+Th)/He dating method. These estimated crystallisation ages are then compared with those of major tectonic and magmatic events in southern Africa, including metasomatic events recorded in the radiogenic isotope systematics of mantle xenoliths, to establish a sequence of diamond-forming metasomatic events in the West Kaapvaal.

The objectives of the study are clear, the quality of the data is good and the approach to obtain diamond crystallisation ages is as robust as it can be given the complexities and limitation of this new dating method (new to date diamonds). I especially praise the Authors for developing a novel method to calculate (U+Th)/He ages which keeps into account He diffusion out of diamond, something not considered in the only previous published effort to get diamond ages by the (U+Th)/He method (Timmermann et al., 2019 cited).

Overall, I am supportive of publication of this work in Nature Communications because it provides a new approach to directly date diamond formation ages. Yet I urge the Authors to carefully consider my comments below. Some are mainly directed to improve the impact and clarity of this work, others are essentially “words of caution” because some of the conclusions are not fully supported by the results or sounds quite speculative.

1) The introductory section, while well framed, overlooks an important topic, that is the well-established application of radiogenic isotope dating of silicate and sulfide inclusions in diamonds to obtain diamond formation ages, including the current debate regarding the meaning of these mineral inclusion ages. In this sense, direct dating of HDF-rich diamonds can provide a means to circumvent this debate and get diamond crystallisation ages with potentially fewer assumptions than for mineral inclusion dating. Hence, I believe the scope and impact of this work could be made broader by including this aspect of diamond dating studies rather than focussing exclusively on the importance of dating HDF-rich diamonds.

Done; we integrated this issue in the introduction (lines 32-46, 49-54, 82-85).

2) Pooling of diamond data for the Kimberley (De Beers pool) and Finsch mines is risky. Kimberley and Finsch are ~100 km apart and they sample a very different mantle column based on existing mantle xenolith and xenocryst data (e.g., Griffin et al., 2003 Lithos, cited). Their different emplacement ages underpin a significant modification of the lithospheric mantle underneath this region between 118 and 85 Ma. In other words, the De Beers diamonds could reflect a different event than the Finsch diamonds, and this should be recognised in the revised manuscript.

The similarity in composition between saline HDFs from the southwest Kaapvaal, including samples from Finsch, De Beers Pool and Koffiefontein, and their host diamonds low nitrogen aggregation (~100% A-centers; Izraeli et al., EPSL, 2001; Weiss et al., EPSL, 2018; Weiss and Goldstein, MIP, 2018; Weiss unpublished data; Timmerman et al., CG and MIP, 2018), clearly indicate a genetic relation and formation during a restricted time period. This is further strengthened by the U-Th/He errorchron for saline HDF-bearing diamonds (Fig. 2c), which gives an age of 90 ± 42 Ma. The reviewer mentioned correctly that the metasomatism in Finsch and De Beers Pool may be related to different events. But in this case, each of the events was short and together they represent few metasomatic events by very similar saline fluids, as part of continuous saline metasomatism that is restricted by time shortly before the Finsch eruption and until the De Beers Pool kimberlites eruptions. In our opinion, this may be regarded as a single continuous metasomatic event by saline HDFs. This is now mentioned in the discussion (lines 263-266).

3) The maximum diffusion coefficient D_{He} calculated by the Authors is based on the assumption that the oldest diamond-forming event in the Kaapvaal was coeval (or postdated) the final assembly of this craton at ~2.6 Ga (L143 and thereabouts). Yet, there are diamonds with 'kimberlitic' features in the Witwatersrand conglomerates which must be older than the conglomerate deposition age of ~2.9 Ga. I would rather use an upper bound of 3.2-3.3 Ga (see Smart et al., 2016 Nature Geo) to calculate maximum D_{He} .

Done; we calculate the maximum diffusion of He based on 3.3 Ga (the maximum age for monocrySTALLINE diamonds in the Kaapvaal), which gives $D \leq 1.93 \times 10^{-19}$ cm²/s (for $^3\text{He}/^4\text{He}_0 \approx 8$ Ra). This is now included in the discussion (lines 230-232). However, we further argue (as in the previous version of the MS) that HDF-bearing diamonds are unlikely to survive the thermal and tectonic history before craton stabilization at ~2.6 Ga, and this implies an upper limit of $\sim 1.8 \times 10^{-19}$ cm²/s for the He diffusion.

4) I am sorry to say that I find the correlation between the Proterozoic diamond forming event and the Namaqua-Natal orogeny highly speculative (L155-160 and thereabouts). There are multiple events occurring in and around the Kaapvaal between >2.6 Ga and 750 Ma (i.e. the permissible ages of diamond formation based on He isotope systematics) beyond the Namaqua-Natal orogeny, including the Bushveld LIP and the Kibaran orogeny just to mention a couple. So, why should this diamond age specifically reflect the Namaqua-Natal orogeny? Are there other diamond formation ages based on mineral inclusions at Finsch which could provide

additional support to a link between diamond formation and the N-N orogeny? The Authors employ Ar-Ar ages of mantle-derived phlogopite to support the conclusion of mantle metasomatism underneath the West Kaapvaal during the N-N orogeny. However, Ar/Ar dating of mantle phlogopite is known to provide spurious ages (Phillips and Onstott, 1988 Geology; Phillips, 1991 Chem Geol; Giuliani et al., 2012 IKC Ext Abstr).

Done. The text has been changed accordingly to give the option for connection between carbonatite metasomatism and the various large tectonic and volcanic events in the Kaapvaal (lines 247-254), and Ar-Ar ages of phlogopite were removed from the text and figures.

5) More broadly, the West Kaapvaal has experienced an endless sequence of tectonic and magmatic events, which may (or may not) have left permanent metasomatic 'scars' on the underlying lithospheric mantle. While the Authors clearly demonstrate a link between different diamond-forming events, mantle fluid compositions and potentially ages, the sequence of mantle fluid infiltration they present sounds too 'definitive'. For example, there is ample evidence of silicate melt metasomatism associated with Bushveld and Karoo (Hoal, 2003 Lithos; Giuliani et al., 2014 EPSL) yet they did not observe a link between formation of silicic HDF-bearing diamond and these major events (maybe because these were diamond-destructive events or simply because of the limited size of the dataset). In simple words, diamonds alone cannot constrain the metasomatic history of the Kaapvaal and a more integrated approach should also include mantle xenoliths and magmatic events. In other words, I would tone down the last part of the discussion and focus the conclusions of this work on the new methodology here developed and its applicability to diamonds elsewhere.

Done; the text has been changed and conclusions toned down.

Also, thank you for mentioning the metasomatism related to the Bushveld and Karoo, which we now bring as an example for specific silicic events that are different from the silicic HDF event that we constrain in the present study (lines 256-259).

6) L172-175. This temporal relationship between formation of diamonds containing saline HDF and Mesozoic kimberlite (plus lamproite) magmatism is interesting. However, I shall note that kimberlite metasomatism of the lithospheric mantle in the western Kaapvaal produces carbonate-rich mantle xenoliths (e.g., Dawson et al., 2001 CMP; Soltys et al., 2016 Lithos; Fitzpayne et al., 2018 Mineral Petrol) with only limited evidence of addition of Cl and Na (e.g., Giuliani et al., 2012 Geology; Giuliani et al., 2013 CMP). So, while the temporal correlation holds, the genetic implications of this correlation deserve further work, which could be also represent a further research avenue stemming out of this important study.

Done; we rephrased the text accordingly (lines the text 271-276). We also included the similarity between in $3\text{He}/4\text{He}$ of saline HDFs and Wesselton kimberlite (as suggested by the reviewer below), to emphasize the possible relation between the two in addition to the observed temporal correlation.

7) L176-180. I must strongly disagree with this statement and in fact with the article of Yaxley et

al. which the Authors refer to here. Kimberlites are sub-lithospheric melts (Smith, 1983 Nature; Torsvik et al., 2010 Nature; Stagno et al., 2013 Nature; Stamm and Schmidt, 2018 EPSL; Pearson et al., 2019 Elements; Giuliani et al., 2020 Sci Adv; etc etc) and their only connection with the cratonic lithosphere is due to the deep melt extraction depths in the asthenosphere provided by thick cratonic roots. Kimberlites do not derive from previously metasomatised SCLM as argued originally by Dan McKenzie (Tainton and Mckenzie, 1994 J Pet) based on their extrapolation of a petrological model that is only valid for basalts and similar melts, not kimberlites. This is a fundamental mistake that has been perpetuated for decades and which has only recently been finally clarified. The isotope geochemistry of kimberlites and link to mantle plumes require a sub-lithospheric source for kimberlites (see additional arguments in the references above).

Done; the text has been changed accordingly and the citation to Yaxley et al. was taken out.

8) It should be clearly explained how (U+Th)/He ratios were calculated because these element concentrations were measured in very different ways (LA-ICPMS vs bulk crushing + static MS). If I recall correctly one of the early works of Weiss et al. positively compared trace element concentrations of HDF-rich diamond obtained by LA-ICPMS and bulk-combustion (or a similar bulk method). I recommend reporting these pieces of information and demonstrate that U and Th concentrations by LA-ICPMS are representative of the bulk of fluid inclusions. This will make the methodology more sound to the reader.

Done; this issue is now included in the Results section – ‘The significance of microinclusion homogeneity.’, which also refers to a new Extended Data Figure 2. The section demonstrates the compositional homogeneity of the inclusions in a diamond, and the similarity of bulk estimations between average LA-ICP-MS analyses and INAA analyses of two HDF-bearing diamonds previously analyzed by Schrauder et al., GCA, 1996 and Rege et al., Lithos, 2010, suggesting that the bulk U and Th composition of a diamond can be closely represented by averaging a few LA-ICP-MS analyses of different microinclusions-bearing zones in a diamond.

Minor comments:

L16-17: This statement might be a bit too strong. At the end of the day, dating of mineral inclusions in diamonds essentially provides ages of diamond-producing metasomatic events in the lithospheric mantle. The novelty of this work is rather that (U-Th)/He dating may be a direct rather than indirect means to date diamond crystallisation events.

Done; changed to ‘However, until now, direct constraints on the timing of such events have not been available.’ (lines 15-16).

L24-25. I would rather say that the ages of these diamond-forming events are coveal with those of major tectonic events recorded in the crust. Also note that the link between mantle metasomatism and tectonic events in previous studies of mantle xenoliths is based on model ages (Lazarov et al.) and Ar/Ar dating (Hopp et al., both cited) where Ar/Ar dating of mantle

phlogopite is well known to provide spurious ages (Phillips and Onstott, 1988 Geology; Phillips, 1991 Chem Geol). So these links are not very robust.

Done; we have changed the sentence – ‘...and are likely coeval with major surface tectonic events (e.g. the Damara and Namaqua–Natal orogenies).’ (lines 22-24).

L34. Some of these refs are probably not ideal in this context (e.g., Carlson and Irvine 1994; Hawkworth et al 1990; Mysen 1983). There are much more recent references with a strong focus on the composition of mantle fluids, both from my group (e.g., Giuliani et al., 2012 Geology; Giuliani et al., 2015 Nature Comms) and others (e.g., see recent papers in Lithos by Jurgen Konzett; the various papers by Maria-Luce Frezzotti on mantle fluids, e.g., 2012 EPSL; 2010 GCA; and the 2001 review of Anderson and Neumann in Lithos).

Done; removed - Carlson and Irvine, 1994; Hawkworth et al., 1990; Mysen, 1983; added - Giuliani et al., 2012; Frezzotti et al., GCA, 2010.

L75. This $3\text{He}/4\text{He}$ range is very similar to that of magmatic fluids in olivine from the Wesselton kimberlite (Giuliani et al., 2020 EGU abstract; attached), which is part of the Kimberley kimberlites where the De Beers Pool diamonds were sourced.

Thank you. This information is very interesting and we used it in the revised manuscript to stress the apparent possible relation between saline HDFs and kimberlites, in addition to their coeval timing. The range of $3\text{He}/4\text{He}$ range of the Wesselton kimberlite is now also included in Fig. 1.

L93. Please include references for these emplacement ages.

Done; added – e.g. Allsopp et al., Kimberlites and related rocks, 1989; Smith et al., Trans Geol Soc S Afr, 1985; and the review paper by Field et al., Ore Geology Reviews, 2008.

L98. I shall also note that southern African Cretaceous kimberlites (Kimberley, but not Finsch, which is an olivine lamproite) feature a strong HIMU component (e.g., Smith, 1983 Nature; Collerson et al., 2010 Phys Chem Earth). HIMU OIBs (Hanyu and Kaneoka, 1997 Nature) and the HIMU-like Wesselton kimberlite (Giuliani EGU abstract) both contains low $3\text{He}/4\text{He}$ similar to these diamonds, which might further support the occurrence of recycled crustal material in the mantle source of these diamond-forming fluids.

Done; see the answer to the comment above – (L75).

L123. based on diamond of which size?

Done; for a diamond radius of 2.5 mm.

This information was mentioned in the diffusion supplementary information of the previous manuscript version; now it is integrated into the diffusion model part within the results section of the main text (line 144-145), as well as in the methods section.

L166. I would not use the term "control" here given the abundant evidence of metasomatism by

carbonate-rich and silicate melts in the Kaapvaal during the Mesozoic recorded in mantle xenolith studies (some examples in articles from my group just to give an idea)

Considering the available data; 25 of 26 HDF-bearing diamonds from De Beers Pool, Koffiefontein and Finsch that have low-aggregated nitrogen are characterized by saline compositions. We therefore hold to our opinion that Mesozoic metasomatism in this lithosphere province was controlled by saline fluids. However, to comply with this comment, we modified the sentence to be more specific to metasomatism that led to diamond formation (i.e. ‘...highly saline fluids, controlled the metasomatism that also led to diamond formation in the Kaapvaal CLM during the Cretaceous’; line 259-260).

Figure 1. I guess the Authors only compiled representative He isotope data for mantle xenoliths given that there are many more studies available than those in the "data compilation" file (e.g., Barry et al., 2015 Lithos; Dunai and Baur, 1995 GCA; Patterson et al., 1994 GCA; Porcelli et al., 1986 GCA and 1992 CMP; Yamamoto et al., 2004 Chem Geol).

Not exactly; Dunai and Baur, 1995 GCA; Patterson et al., 1994 GCA; Porcelli et al., 1986 GCA and 1992 CMP; Yamamoto et al., 2004 Chem Geol, and other studies are intentionally not included in the compilation as they used heating and fusion to release the He rather than crushing. The compilation therefore presents only crushing data, similar to the extraction methods for releasing and measuring the He in HDF-bearing diamonds. We included this info in the text and the compilation (Supplementary data Table 1). Barry et al., Lithos, 2015, is now included in the compilation.

Figure 1. My EGU2020 abstract (n. 5267) also report a $^3\text{He}/^4\text{He}$ range of 1.6-3.7 R/Ra for fluid-inclusion rich olivine grains for the Wesselton kimberlite. It could perhaps be helpful to include these data as representative of the He isotope composition of the kimberlite host (at least for the De Beers Pool diamonds).

Done; see the answer to the comment above – (L75).

L462-463. This method (i.e. crushing of inner diamond fragments) avoids potential issues with alpha-implantation and ejection. I would stress this somewhere in the main text.

Done; we stress it now as follows in lines 127-131, ‘The prime source of He in HDF-bearing diamonds is the Microinclusions (Broadley et al., 2018b; Burgess et al., 1998; Timmerman et al., 2019). Additional possible sources of He from the diamond lattice are negligible, or can be avoided by crushing release of He from an inner fragment of deep mined diamonds (as discussed in details by Timmerman et al., 2019, and here in the Supplementary Information – “The budget of He in HDF-bearing diamonds”).’.

Additional minor suggestions and stylistic comments are included in the attached pdf.

Done; the few suggestions we did not agree with are listed and explained below.

Line 70: “including the so-called ‘low- $^3\text{He}/^4\text{He}$ ’ OIBs ($^3\text{He}/^4\text{He} < 7 \text{ Ra}$)^{34,35}.”

We think it is important to mention those in particular as they are related to subduction. Something that we later use as supporting evidence for the origin of the saline HDFs. We therefore decide to leave it as it is.

Line 125: “this part seems to be redundant unless I missed something”

We think it is actually important to mention this because it is necessary for explaining the change in (U-Th)/He model ages with time.

Line 150: “which shows a protracted history of depletion and followed by multiple metasomatic events”

Thank you, we considered it as an alternative but decided to leave the sentence as it is.

Line 175: “Finsch is an olivine lamproite (previously orangeite) not a kimberlite”

True, Finsch is known as Group II kimberlite (orangeite), and we include now a sentence explaining that we use the term kimberlite in general including Finsch (line 124-125).

Overall this is a nice and important contribution which advances the toolkit available to understand mantle processes. Hence, I reiterate my support towards its publication.

I hope the Authors will find my comments helpful and I am available for any clarification.

Andrea Giuliani

Zurich, 12-Aug-2020

Reviewer #2 (Remarks to the Author):

Diamonds provide unique opportunities to explore the history of processes occurring in the mantle. Dating diamonds, particularly zoned crystals or deposits with multiple generations of diamond formation is one of the most important challenges in the field. Dating fluid inclusions with (U-Th)/He methods is therefore very exciting, particularly if it can be related to different fluid compositions and defect concentrations in host diamonds. The study has the potential to be of sufficient interest to be considered for Nature Communications, but the quality of the execution and the significance of the findings needs to match the potential of the method.

Overall recommendation – I do not recommend that the paper is published in Nature

Communications in its current form for the following reasons.

1. Although the concept is simple (and good) the detail is extremely important for this paper. I found it very hard to follow the detail in a paper of this style, and the logic of the story was not always clear to me. I wasted a lot of time poring over the supplementary information, extended data and previous papers by the authors. I think the paper would be much better suited a longer paper format where the methods, materials and logical flow of the arguments would be clearer.

We have made major attempts to deal with these issues and we believe that the current version does so. We rewrote large parts of the paper and integrated a lot of details that were previously in the supplementary information to the Results sections within the main text and also the Methods section.

The section about the possible sources contributing to the budget of He in an HDF-bearing diamond, which was also discussed in detail by Timmerman et al., CG, 2019 (mentioned in the text, lines 127-131), remains in the supplementary information.

We believe the detail, logic, and story given in the current manuscript are appropriate for publication and understandable to a reader.

2. I am not convinced by the novelty compared with previous papers. The main new contribution is the diffusion modelling, which is the aspect I am least convinced by.

We have emphasized the importance of the diffusion model, which we maintain is novel, and constraints that it places on the diffusion of He in HDF-bearing diamonds, which is clearly important for meaningful (U-Th)/He geochronology (Fig. 3-5 and Extended Data Figure 1). We maintain that this exercise resulted in a major step forward towards unraveling the age of HDF-bearing diamonds and for constraining the timing of the metasomatic events in which they formed.

We are not certain how to address the second part of the comment.

3. I found the description of the analysis and diffusion modelling to be confusing. At one point it is stated that He was analysed in “inner” microinclusion-bearing portions of the sample. As far as I could understand only gas from the microinclusions was analysed, not He from the diamond lattice.

It is correct that we analyzed He in the microinclusions; we intentionally avoid contributions from the diamond lattice by crushing rather than heating/burning the diamond. Combining the analyses results with the diffusion modeling constrains the possible time range of diamond’s crystallization and metasomatic event by providing minimum and maximum model ages. Regarding the diffusion model, we have made major efforts to clarify what we did, the results, and the implications.

There is also a discussion of where inclusions are in the samples, but no images. The modelling appears only to apply to a homogeneous distribution of inclusions (if I understand it correctly)

but images in Weiss et al 2018 and Weiss and Goldstein 2018 of the same samples, clearly show variable core-rim relationships. Thus the distance between inclusion-rich zone and diamond surface would be very important (rather than the overall radius of the diamond).

Yes, images for the different diamonds analyzed in the present study are found in Weiss et al., EPSL, 2018 and Weiss and Goldstein, MIP, 2018. We refer to these two studies and don't see a place to include the images in the present study as well.

It is important to note that the diffusion model does not only apply to diamonds with homogeneous inclusion density, because an average of few LA-ICP-MS analyses in different zones closely represent the inclusion density variation and provides a good estimation of the bulk U and Th of the diamond. The problem arises in diamonds that contain internal microinclusion-bearing and overgrown microinclusion-free zones, which cannot be entirely separated for He analyses. In these cases, a shortfall of He compared to U and Th (measured at the microinclusions-bearing zone solely) may result in an apparent high (U-Th)/He ratios. This was the case for two diamonds from Finsch, which therefore were not used for age calculations. These issues are now included in the text – ‘The significance of microinclusion homogeneity’. (lines 171-199).

On top of these issues, the diffusivity of He (and the assumed equivalence of 3He and 4He) are so poorly known I would question the value of the modelling results.

Regarding the diffusivity of He, this is true and we modeled it for different D-values. One of our results is to estimate the maximum D-value. We believe both the model and the constraints are important contributions. It is true that one of the model assumptions is that $D(3\text{He}) \approx D(4\text{He})$; this is stated in the text (Methods section). Our opinion is that this is the most reasonable and appropriate approach for the subject at this point, given the absence of experimental constraints about possible differences between 4He and 3He diffusivities in diamonds.

Finally, there is a relationship between measurable quantities (N concentration and aggregation), time (the output of the paper) and temperature (a major controlling factor in He diffusivity), so an opportunity for a more sophisticated iterative approach has been missed.

We appreciate the comment, and we will consider it for future studies.

Other issues.

1. No uncertainties are given in extended data table 1, and no uncertainties are propagated or discussed. There are no error bars in fig 1a, 1b or 2a, but there are in 2b and 2c. The best fit in 2c is not an isochron, as only 3/5 (3/6 depending on selection) lie on the line within the stated errors. It is an errorchron. This must be explained in any reworking of the paper. Either the errors are much larger than stated, or the broad “isochron-like trend” arises from process other than radioactive decay.

Done; uncertainties for %B-centers, Th and U are now included in Extended data table 1. Error bars were included in the figures in the previous version of the MS and all were propagated

where needed; for most of the data points, the 4He content and $3\text{He}/4\text{He}$ errors are smaller than the size of the symbols. We added errors of %B-centers to Fig. 2a, as maximum possible errors, and errors for Timmerman et al., MIP, 2018 to Fig. 1a, 1b and 2a.

Regarding the comment that the errorchron arises from processes other than radioactive decay, while this is not an “isochron” sensu stricto in that the data do not fall precisely on a line, we maintain that it is likely not a coincidence that the errorchron age coincides with the eruption age of the kimberlite, and we have every reason to believe that the trend does reflect radioactive decay. Nevertheless, we adopt the suggestion by the reviewer and now call the $4\text{He}/3\text{He}$ vs. $(\text{U}+\text{Th})/3\text{He}$ trend observed for the saline HDF-bearing diamonds – ‘errorchron’.

2. The authors are well aware that N aggregation is affected by all of time, temperature and total N concentration, so plotting $4\text{He}/3\text{He}$ vs N aggregation is a strange choice that is never justified in the text.

We are not sure why the reviewer writes this, as we are plotting two independent variables that are a function of time, i.e. $4\text{He}/3\text{He}$ vs N aggregation, this approach is clearly justified to evaluate the effects of time on process. Moreover, the positive correlation observed between them illustrates the impact of time on both. The N concentration in the diamonds, which is overall similar, and possible temperature differences, cannot explain the differences observed. This is discussed in detail in Weiss et al., EPSL, 2018 and Weiss and Goldstein, MIP, 2018, and we don’t see a justified reason to discuss it here again. Moreover, if the important term controlling the N aggregation in the studied diamond was temperature, rather than time, it would unlikely to appear as a positive correlation with $4\text{He}/3\text{He}$ ratios (and could possibly even form a negative correlation).

3. In Extended data figure 1, why does the function for 50 Ma look so different from all the other times? It looks very strange. Are you sure it is not an error?

Thank you for this comment; this was a confusion between 0 Ma and 50 Ma which is now corrected.

In summary, the paper has the potential to be an exciting and valuable contribution to the field. However, there are a lot of potential problems with the data and the modelling, so the paper needs to be written in a way that the reader can follow more exactly what has been done so that the uncertainties and limitations are transparent. The short paper format is not suitable for the depth of treatment that is required.

REVIEWERS' COMMENTS

Reviewer #2 (Remarks to the Author):

I believe the Authors have carefully addressed all my comments and the manuscript is considerably improved in my opinion. The results section includes abundant additional details, which make the logic of this work more robust. Similarly, the more speculative parts of the discussion have been removed and the section is overall more sound. I can only reiterate my support for the publication of this important manuscript.

Here is my last (pedantic) suggested fix:

Line 14: I would replace "their entrapment" with "entrapment of these fluids". It is not clear from the statement that "their" refer to fluids.

Best wishes,
Andrea Giuliani
Zurich, 10-Feb-2021

Response to reviewers' comments (second-round): Helium in diamonds unravels over a billion years of craton metasomatism, by Yaakov Weiss, Yael Kiro, Cornelia Class, Gisela Winckler, Jeff W. Harris, and Steven L. Goldstein.

We thank the reviewers for all their comments and we acknowledge that these considerably improved the manuscript.

The reviewers' comments are italicized below and our responses are integrated within the text in blue non-italic text.

REVIEWERS' COMMENTS

Reviewer #2 (Remarks to the Author):

I believe the Authors have carefully addressed all my comments and the manuscript is considerably improved in my opinion. The results section includes abundant additional details, which make the logic of this work more robust. Similarly, the more speculative parts of the discussion have been removed and the section is overall more sound. I can only reiterate my support for the publication of this important manuscript.

Here is my last (pedantic) suggested fix:

Line 14: I would replace "their entrapment" with "entrapment of these fluids". It is not clear from the statement that "their" refer to fluids.

Done.

*Best wishes,
Andrea Giuliani
Zurich, 10-Feb-2021*

Thank you.

Yaakov Weiss